# Hierarchical Relational Learning for Few-Shot Knowledge Graph Completion

**Han Wu[1], Jie Yin[1], Bala Rajaratnam[1,2] & Jianyuan Guo[1]**
[1]The University of Sydney, [2]University of California, Davis
{han.wu,jie.yin,bala.rajaratnam,jguo5172}@sydney.edu.au

## Abstract

Knowledge graphs (KGs) are powerful in terms of their inference abilities, but are also notorious for their incompleteness and long-tail distribution of relations. To address these challenges and expand the coverage of KGs, few-shot KG completion aims to make predictions for triplets involving novel relations when only a few training triplets are provided as reference. Previous methods have focused on designing local neighbor aggregators to learn entity-level information and/or imposing a potentially invalid sequential dependency assumption at the triplet level to learn meta relation information. However, pairwise triplet-level interactions and context-level relational information have been largely overlooked for learning meta representations of few-shot relations. In this paper, we propose a hierarchical relational learning method (HiRe) for few-shot KG completion. By jointly capturing three levels of relational information (entity-level, triplet-level and context-level), HiRe can effectively learn and refine meta representations of few-shot relations, and thus generalize well to new unseen relations. Extensive experiments on benchmark datasets validate the superiority of HiRe over state-of-the-art methods. The code can be found in https://github.com/alexhw15/HiRe.git.

## 1 Introduction

Knowledge graphs (KGs) comprise a collection of factual triplets, $(h, r, t)$, where each triplet expresses the relationship $r$ between a head entity $h$ and a tail entity $t$. Large-scale KGs (Vrandečić & Krötzsch, 2014; Mitchell et al., 2018; Suchanek et al., 2007; Bollacker et al., 2008) can provide powerful inference capabilities for many intelligent applications, including question answering (Yao & Van Durme, 2014), web search (Eder, 2012) and recommendation systems (Wang et al., 2019).

As KGs are often built semi-automatically from unstructured data, real-world KGs are far from complete and suffer from the notorious long-tail problem — a considerable proportion of relations are associated with only very few triplets. As a result, the performance of current KG completion methods significantly degrades when predicting relations with a limited number (few-shot) of training triplets. To tackle this challenge, few-shot KG completion methods have been proposed including GMatching (Xiong et al., 2018), MetaR(Chen et al., 2019), FSRL(Zhang et al., 2020), FAAN (Sheng et al., 2020) and GANA (Niu et al., 2021). These methods focus on predicting the missing tail entity $t$ for *query triplets* by learning from only $\mathcal{K}$ *reference triplets* about a target relation $r$.

Given a target relation $r$ and $\mathcal{K}$ reference triplets, $\mathcal{K}$-shot KG completion aims to correctly predict the tail entity $t$ for each query triplet $(h, r, ?)$ using the generalizable knowledge learned from reference triplets. Thus, the crucial aspect of few-shot KG completion is to learn the meta representation of each few-shot relation from a limited amount of reference triplets that can generalize to novel relations. To facilitate the learning of meta relation representations, we identify three levels of relational information (see Figure 1). (1) At the **context level**, each reference triplet is closely related to its wider contexts, providing crucial evidence for enriching entity and relation embeddings. (2) At the **triplet level**, capturing the commonality among limited reference triplets is essential for learning meta relation representations. (3) At the **entity level**, the learned meta relation representations should well generalize to unseen query triplets.

Current few-shot KG methods have, however, focused on designing local neighbor aggregators to learn entity-level information, and/or imposing a sequential assumption at the triplet level to learn meta relation information (See Table 1). The potential of leveraging pairwise triplet-level interactions and context-level relational information has been largely unexplored.

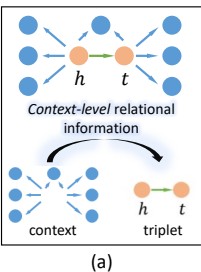 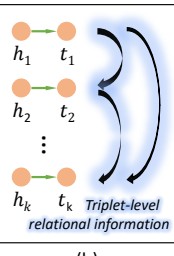 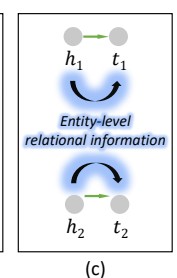

|         (a)         |         (b)          |          (c)          |

Figure 1: Three levels of relational information: (a) Context-level, (b) Triplet-level, (c) Entity-level.

| Methods | Entity-level | Triplet-level | | Context-level |
|---|---|---|---|---|
| | | Seq. | Pair. | |
| GMatching | ✓ | ✗ | ✗ | ✗ |
| MetaR | ✓ | ✗ | ✗ | ✗ |
| FSRL | ✓ | ✓ | ✗ | ✗ |
| FAAN | ✓ | ✓ | ✗ | ✗ |
| GANA | ✓ | ✓ | ✗ | ✗ |
| **HiRe (ours)** | ✓ | ✗ | ✓ | ✓ |

Table 1: Summary of few-shot KG completion methods based on different levels of relational information used.

In this paper, we propose a **Hi**erarchical **Re**lational learning framework (**HiRe**) for few-shot KG completion. HiRe jointly models three levels of relational information (entity-level, triplet-level, and context-level) within each few-shot task as mutually reinforcing sources of information to generalize to few-shot relations. Here, "hierarchical" references relational learning performed at three different levels of granularity. Specifically, we make the following contributions:

- We propose a **contrastive learning based context-level relational learning** method to learn expressive entity/relation embeddings by modeling correlations between the target triplet and its true/false contexts. We argue that a triplet itself has a close relationship with its true context. Thus, we take a contrastive approach — a given triplet should be pulled close to its true context, but pushed apart from its false contexts — to learn better entity embeddings.
- We propose a **transformer based meta relation learner (MRL)** to learn generalizable meta relation representations. Our proposed MRL is capable of capturing pairwise interactions among reference triplets, while preserving the permutation-invariance property and being insensitive to the size of the reference set.
- We devise a **meta representation based embedding learner** named **MTransD** that constrains the learned meta relation representations to hold between unseen query triplets, enabling better generalization to novel relations.

Lastly, we adopt a model agnostic meta learning (MAML) based training strategy (Finn et al., 2017) to optimize HiRe on each meta task within a unified framework. By performing relational learning at three granular levels, HiRe offers significant advantages for extracting expressive meta relation representations and improving model generalizability for few-shot KG completion. Extensive experiments on two benchmark datasets validate the superiority of HiRe over state-of-the-art methods.

## 2 RELATED WORK

### 2.1 RELATIONAL LEARNING IN KNOWLEDGE GRAPHS

KG completion methods utilize relational information available in KGs to learn a unified low-dimensional embedding space for the input triplets. TransE (Bordes et al., 2013) is the first to use relation $r$ as a translation for learning an embedding space, *i.e.*, $\mathbf{h} + \mathbf{r} \approx \mathbf{t}$ for triplet $(h, r, t)$. A scoring function is then used to measure the quality of the translation and to learn a unified embedding space. TransH (Wang et al., 2014) and TransR (Lin et al., 2015) further model relation-specific information for learning an embedding space. ComplEx (Trouillon et al., 2016), RotatE (Sun et al., 2019b), and ComplEx-N3 (Lacroix et al., 2018) improve the modeling of relation patterns in a vector/complex space. ConvE (Dettmers et al., 2018) and ConvKB (Nguyen et al., 2018) employ convolution operators to enhance entity/relation embedding learning. However, these methods require a large number of triplets for each relation to learn a unified embedding space. Their performance significantly degrades at few-shot settings, where only very few triplets are available for each relation.

### 2.2 FEW-SHOT KG COMPLETION

Existing few-shot KG completion methods can be grouped into two main categories: (1) Metric learning based methods: GMatching (Xiong et al., 2018) is the first work to formulate few-shot (one-shot) KG completion. GMatching consists of two parts: a neighbor encoder that aggregates one-hop neighbors of any given entity, and a matching processor that compares similarity between query and reference entity pairs. FSRL (Zhang et al., 2020) relaxes the setting to more shots and explores how to integrate the information learned from multiple reference triplets. FAAN (Sheng et al., 2020) proposes a dynamic attention mechanism for designing one-hop neighbor aggregators.

(2) Meta learning based methods: MetaR (Chen et al., 2019) learns to transfer relation-specific meta information, but it simply generates meta relation representations by averaging the representations of all reference triplets. GANA (Niu et al., 2021) puts more emphasis on neighboring information and accordingly proposes a gated and attentive neighbor aggregator.

Despite excellent empirical performance, the aforementioned methods suffer from two major limitations. First, they focus on designing local neighbor aggregators to learn entity-level information. Second, they impose a potentially invalid sequential dependency assumption and utilize recurrent processors (*i.e.*, LSTMs (Hochreiter & Schmidhuber, 1996)) to learn meta relation representations. Thus, current methods fail to capture pairwise triplet-level interactions and context-level relational information. Our work is proposed to fill this important research gap in the literature.

### 2.3 Contrastive Learning on Graphs

As a self-supervised learning scheme, contrastive learning follows the instance discrimination principle that pairs instances according to whether they are derived from the same instance (*i.e.*, positive pairs) or not (*i.e.*, negative pairs) (Hadsell et al., 2006; Dosovitskiy et al., 2014).

Contrastive methods have been recently proposed to learn expressive node embeddings on graphs. In general, these methods train a graph encoder that produces node embeddings and a discriminator that distinguishes similar node embedding pairs from those dissimilar ones. DGI (Velickovic et al., 2019) trains a node encoder to maximize mutual information between patch representations and high-level graph summaries. InfoGraph (Sun et al., 2019a) contrasts a global graph representation with substructure representations. (Hassani & Khasahmadi, 2020) propose to contrast encodings from one-hop neighbors and a graph diffusion. GCC (Qiu et al., 2020) is a pre-training framework that leverages contrastive learning to capture structural properties across multiple networks.

In KGs, we note that positive and negative pairs naturally exist in the few-shot KG completion problem. However, the potential of contrastive learning in this task is under-explored. In our work, we adopt the idea of contrastive learning at the context level to capture correlations between a target triplet and its wider context. This enables enriching the expressiveness of entity embeddings and improving model generalization for few-shot KG completion. To the best of our knowledge, we are the first to integrate contrastive learning with KG embedding learning for few-shot KG completion.

## 3 Problem Formulation

In this section, we formally define the few-shot KG completion task and problem setting. The notations used in the paper can be found in Appendix A.

**Definition 1** *Knowledge Graph* $\mathcal{G}$. A knowledge graph (KG) can be denoted as $\mathcal{G} = \{\mathcal{E}, \mathcal{R}, \mathcal{TP}\}$. $\mathcal{E}$ and $\mathcal{R}$ are the entity set and the relation set, respectively. $\mathcal{TP} = \{(h, r, t) \in \mathcal{E} \times \mathcal{R} \times \mathcal{E}\}$ denotes the set of all triplets in the knowledge graph.

**Definition 2** *Few-shot KG Completion*. Given (i) a KG $\mathcal{G} = \{\mathcal{E}, \mathcal{R}, \mathcal{TP}\}$, (ii) a *reference set* $\mathcal{S}_r = \{(h_i, t_i) \in \mathcal{E} \times \mathcal{E} | \exists r, \text{ s.t. } (h_i, r, t_i) \in \mathcal{TP}\}$ that corresponds to a given relation $r \in \mathcal{R}$, where $|\mathcal{S}_r| = \mathcal{K}$, and (iii) a *query set* $\mathcal{Q}_r = \{(\tilde{h}_j, r, ?)\}$ that also corresponds to relation $r$, the $\mathcal{K}$-shot KG completion task aims to predict the true tail entity for each triplet from $\mathcal{Q}_r$ based on the knowledge learned from $\mathcal{G}$ and $\mathcal{S}_r$. For each query triplet $(\tilde{h}_j, r, ?) \in \mathcal{Q}_r$, given a set of candidates $\mathcal{C}_{\tilde{h}_j, r}$ for the missing tail entity, the goal is to rank the true tail entity highest among $\mathcal{C}_{\tilde{h}_j, r}$.

As the definition states, few-shot KG completion is a relation-specific task. The goal of a few-shot KG completion model is to correctly make predictions for new triplets involving relation $r$ when only a few triplets associated with $r$ are available. Therefore, the training process is based on the unit of tasks, where each task is to predict for new triplets associated with a given relation $r$, denoted as $\mathcal{T}_r$. Each meta training task $\mathcal{T}_r$ corresponds to a given relation $r$ and is composed of a reference set $\mathcal{S}_r$ and a query set $\mathcal{Q}_r$, *i.e.* $\mathcal{T}_r = \{\mathcal{S}_r, \mathcal{Q}_r\}$:

$$\mathcal{S}_r = \{(h_1, t_1), (h_2, t_2), ..., (h_\mathcal{K}, t_\mathcal{K})\}, \tag{1}$$

$$\mathcal{Q}_r = \{(\tilde{h}_1, \mathcal{C}_{\tilde{h}_1, r}), (\tilde{h}_2, \mathcal{C}_{\tilde{h}_2, r}), ..., (\tilde{h}_\mathcal{M}, \mathcal{C}_{\tilde{h}_\mathcal{M}, r})\}, \tag{2}$$

where $\mathcal{M}$ is the size of query set $\mathcal{Q}_r$. Mathematically, the training set can be denoted as $\mathcal{T}_{train} = \{\mathcal{T}_i\}_{i=1}^I$. The test set $\mathcal{T}_{test} = \{\mathcal{T}_j\}_{j=1}^J$ can be similarly denoted. Note that all triplets corresponding to the relations in test set are unseen during training, *i.e.*, $\mathcal{T}_{train} \cap \mathcal{T}_{test} = \varnothing$.

Figure 2: Entity neighborhoods *v.s.* Triplet context. Our method jointly considers the context of the target triplet to enable the identification of crucial information, as highlighted in the right figure.

## 4 THE PROPOSED METHOD

In this section, we present our proposed learning framework in details. As discussed earlier, we identify the research gap in $\mathcal{K}$-shot KG completion, where learning only entity-level relational information and capturing sequential dependencies between reference triplets prevent the model from capturing a more stereoscopic and generalizable representation for the target relation. To fill this gap, we propose to perform three hierarchical levels of relational learning within each meta task (context-level, triplet-level and entity-level) for few-shot KG completion. An overview of our proposed hierarchical relational learning (HiRe) framework can be found in Appendix C.

### 4.1 CONTRASTIVE LEARNING BASED CONTEXT-LEVEL RELATIONAL LEARNING

Given a reference set $\mathcal{S}_r$ with $\mathcal{K}$ training triplets, existing works (*e.g.*, (Xiong et al., 2018; Niu et al., 2021)) seek to learn better head/tail entity embeddings by aggregating information from its respective local neighbors, and then concatenate them as triplet representation. Although treating the neighborhoods of head/tail entity separately is a common practice in homogeneous graphs, we argue that this approach is sub-optimal for few-shot KG completion due to the loss of critical information.

Taking Figure 2 as an example, jointly considering the wider context shared by head/tail entity would reveal crucial information – both "Lionel Messi" and "Sergio Agüero" *playFor* "Argentina's National Team" – for determining whether the relation *workWith* holds between the given entity pair ("Lionel Messi", "Sergio Agüero"). Notably, our statistical analysis further affirms that the triplets on KGs indeed share a significant amount of context information (see Appendix B).

Motivated by this important observation, we propose the idea of jointly considering the neighborhoods of head/tail entity as the context of a given triplet to exploit more meticulous relational information. To imbue the embedding of the target triplet with such contextual information, a contrastive loss is employed by contrasting the triplet with its true context against false ones.

Formally, given a target triplet $(h, r, t)$, we denote its wider context as $\mathcal{C}_{(h,r,t)} = \mathcal{N}_h \cup \mathcal{N}_t$, where $\mathcal{N}_h = \{(r_i, t_i)|(h, r_i, t_i) \in \mathcal{TP}\}$ and $\mathcal{N}_t = \{(r_j, t_j)|(t, r_j, t_j) \in \mathcal{TP}\}$. Our goal is to capture context-level relational information — the correlation between the given triplet $(h, r, t)$ and its true context $\mathcal{C}_{(h,r,t)}$. We further propose a multi-head self-attention (MSA) based context encoder, which models the interactions among the given context $\mathcal{C}_{(h,r,t)}$ and assigns larger weights to more important relation-entity tuples within the context shared by head entity $h$ and tail entity $t$.

Specifically, given a target triplet $(h, r, t)$ and its context $\mathcal{C}_{(h,r,t)}$, each relation-entity tuple $(r_i, t_i) \in \mathcal{C}_{(h,r,t)}$ is first encoded as $\mathbf{re}_i = \mathbf{r}_i \oplus \mathbf{t}_i$, where $\mathbf{r}_i \in \mathbb{R}^d$ and $\mathbf{t}_i \in \mathbb{R}^d$ are the relation and entity embedding, respectively, and $\mathbf{r}_i \oplus \mathbf{t}_i$ indicates the concatenation of two vectors $\mathbf{r}_i$ and $\mathbf{t}_i$. An MSA block is then employed to uncover the underlying relationships within the context and generate context embedding $\mathbf{c}$:

$$\mathbf{c}_0 = [\mathbf{re}_1; \mathbf{re}_2; ...; \mathbf{re}_K], \qquad K = |\mathcal{C}_{(h,r,t)}|, \tag{3}$$

$$\mathbf{c} = \sum_{i=0}^{K} \alpha \cdot \mathbf{re}_i, \qquad \alpha = \text{MSA}(\mathbf{c}_0), \tag{4}$$

where $\mathbf{c}_0$ is the concatenation of the embeddings of all relation-entity tuples and $|x|$ is the size of set $x$. The self-attention scores among all relation-entity tuples from $\mathcal{C}_{(h,r,t)}$ can be computed by Eq. 4. Tuples with higher correlations would be given larger weights and contribute more towards the embedding of $\mathcal{C}_{(h,r,t)}$. The detailed implementation of MSA is given in Appendix D.1.

Additionally, we synthesize a group of false contexts $\{\tilde{\mathcal{C}}_{(h,r,t)i}\}$ by randomly corrupting the relation or entity of each relation-entity tuple $(r_i, t_i) \in \mathcal{C}_{(h,r,t)}$. The embedding of each false context $\tilde{\mathcal{C}}_{(h,r,t)i}$

can be learned via the context encoder as $\tilde{\mathbf{c}}_i$. Then, we use a contrastive loss to pull close the embedding of the target triplet with its true context and to push away from its false contexts. The contrastive loss function is defined as follows:

$$\mathcal{L}_c = -\log \frac{\exp(\mathrm{sim}(\mathbf{h} \oplus \mathbf{t}, \mathbf{c})/\tau)}{\sum_{i=0}^{N} \exp(\mathrm{sim}(\mathbf{h} \oplus \mathbf{t}, \tilde{\mathbf{c}}_i)/\tau)}, \tag{5}$$

where $N$ is the number of false contexts for $(h, r, t)$, $\tau$ denotes the temperature parameter, $\mathbf{h} \oplus \mathbf{t}$ indicates the triplet embedding represented as the concatenation of its head and tail entity embeddings, $\mathrm{sim}(x, y)$ measures the cosine similarity between $x$ and $y$. As such, we can inject context-level knowledge into entity embeddings attending to key elements within the context of the given triplet.

## 4.2 TRANSFORMER BASED TRIPLET-LEVEL RELATIONAL LEARNING

After obtaining the embeddings of all reference triplets, the next focus is to learn meta representation for the target relation $r$. State-of-the-art models (*e.g.*, FSRL (Zhang et al., 2020) and GANA (Niu et al., 2021)) utilize LSTMs for this purpose, which inevitably imposes an unrealistic sequential dependency assumption on reference triplets since LSTMs are designed to model sequence data. However, reference triplets associated with the same relation are not sequentially dependent on each other; the occurrence of one reference triplet does not necessarily lead to other triplets in the reference set. Consequently, these LSTM based models violate two important properties. First, the model should be insensitive to the size of reference set (*i.e.*, few-shot size $\mathcal{K}$). Second, the triplets in the reference set should be permutation-invariant. To address these issues, we resort to modeling complex interactions among reference triplets for learning generalizable meta relational knowledge.

To this end, we propose a transformer based meta relation learner (MRL) that effectively models pairwise interactions among reference triplets to learn meta relation representations that generalize well to new unseen relations. There are two main considerations in our model design. (1) Reference triplets are permutation-invariant; (2) Reference triplets that are more representative should be given larger weights when learning meta relation representation. Inspired by Set Transformer (Lee et al., 2019), we design our MRL based on the idea of set attention block (SAB), which takes a set of objects as input and performs self-attention mechanism between the elements. Therefore, our proposed MRL can model pairwise triplet-triplet interactions within $\mathcal{S}_r$ so as to cultivate the ability to learn generalizable meta representation of the target relation.

Mathematically, given a meta training task $\mathcal{T}_r$ targeting at relation $r$, our proposed MRL takes the head/tail entity pairs from its reference set as input, *i.e.*, $\{(h_i, t_i) \in \mathcal{S}_r\}$. Each reference triplet is encoded as $\mathbf{x}_i = \mathbf{h}_i \oplus \mathbf{t}_i$, where $\mathbf{h}_i \in \mathbb{R}^d$ and $\mathbf{t}_i \in \mathbb{R}^d$ are the embeddings of entity $h_i$ and tail entity $t_i$ with dimension $d$. Note that the same entity embeddings $\mathbf{h}_i$ and $\mathbf{t}_i$ are used here as in Eq. 5.

For all reference triplets associated with the same relation $r$, our proposed MRL aims to capture the commonality among these reference triplets and obtain the meta representation for relation $r$. To comprehensively incorporate triplet-level relational information in the reference set, we leverage an SAB on the embeddings of all reference triplets from $\mathcal{S}_r$ (see the details of SAB in Appendix D.2):

$$\mathbf{X} = [\mathbf{x}_0; \mathbf{x}_1; ...; \mathbf{x}_K], \qquad \mathbf{x}_i \in \mathbb{R}^{2d}, \quad 0 \le i \le \mathcal{K}, \tag{6}$$

$$\mathbf{X}' = \mathrm{SAB}(\mathbf{X}) \in \mathbb{R}^{\mathcal{K} \times 2d}, \tag{7}$$

where $\mathbf{x}_i$ denotes the $i$-th reference triplet. The output of SAB has the same size of the input $\mathbf{X}$, but contains pairwise triplet-triplet interactions among $\mathbf{X}$. The transformed embeddings of reference triplets, $\mathbf{X}'$, are then fed into a two-layer MLP to obtain the meta representation $\mathcal{R}_{\mathcal{T}_r}$, given by

$$\mathcal{R}_{\mathcal{T}_r} = \frac{1}{\mathcal{K}} \sum_{i=1}^{\mathcal{K}} \mathrm{MLP}(\mathbf{X}'), \tag{8}$$

where the meta representation $\mathcal{R}_{\mathcal{T}_r}$ is generated by averaging the transformed embeddings of all reference triplets. This ensures that $\mathcal{R}_{\mathcal{T}_r}$ contains the fused pairwise triplet-triplet interactions among the reference set in a permutation-invariant manner.

## 4.3 META REPRESENTATION BASED ENTITY-LEVEL RELATIONAL LEARNING

A crucial aspect of few-shot KG completion is to warrant the generalizability of the learned meta representation. The learned meta representation $\mathcal{R}_{\mathcal{T}_r}$ should hold between $(\mathbf{h}_i, \mathbf{t}_i)$ if $h_i$ and $t_i$ are

associated with $r$. This motivates us to refine $\mathcal{R}_{\mathcal{T}_r}$ under the constraints of true/false entity pairs. Translational models provide an intuitive solution by using the relation as a translation, enabling to explicitly model and constrain the learning of generalizable meta knowledge at the entity level. Following KG translational models (Bordes et al., 2013; Ji et al., 2015), we design a score function that accounts for the diversity of entities and relations to satisfy such constraints. Our method is referred to as *MTransD* that effectively captures meta translational relationships at the entity level.

Given a target relation $r$ and its corresponding reference set $\mathcal{S}_r$, after obtaining its meta representation $\mathcal{R}_{\mathcal{T}_r}$, we can now calculate a score for each entity pair $(h_i, t_i) \in \mathcal{S}_r$ by projecting the embeddings of head/tail entity into a latent space determined by its corresponding entities and relation simultaneously. Mathematically, the projection process and the score function can be formulated as:

$$\mathbf{h}_{pi} = \mathbf{r}_{pi}\mathbf{h}_{pi}^{\mathsf{T}}\mathbf{h}_i + \mathbf{I}^{m \times n}\mathbf{h}_i, \tag{9}$$

$$\mathbf{t}_{pi} = \mathbf{r}_{pi}\mathbf{t}_{pi}^{\mathsf{T}}\mathbf{t}_i + \mathbf{I}^{m \times n}\mathbf{t}_i, \tag{10}$$

$$\text{score}(h_i, t_i) = ||\mathbf{h}_{pi} + \mathcal{R}_{\mathcal{T}_r} - \mathbf{t}_{pi}||_2, \tag{11}$$

where $||\mathbf{x}||_2$ represents the $\ell_2$ norm of vector $\mathbf{x}$, $\mathbf{h}_i/\mathbf{t}_i$ are the head/tail entity embeddings, $\mathbf{h}_{pi}/\mathbf{t}_{pi}$ are their corresponding projection vectors. $\mathbf{r}_{pi}$ is the projection vector of $\mathcal{R}_{\mathcal{T}_r}$, and $\mathbf{I}^{m \times n}$ is an identity matrix (Ji et al., 2015). In this way, the projection matrices of each head/tail entity are determined by both the entity itself and its associated relation. As a result, the projected tail entity embedding should be closest to the projected head entity embedding after being translated by $\mathcal{R}_{\mathcal{T}_r}$. That is to say, for these triplets associated with relation $r$, the corresponding meta representation $\mathcal{R}_{\mathcal{T}_r}$ should hold between $\mathbf{h}_{pi}$ and $\mathbf{t}_{pi}$ at the entity level in the projection space. Considering the entire reference set, we can further define a loss function as follows:

$$\mathcal{L}(\mathcal{S}_r) = \sum\nolimits_{(h_i, t_i) \in \mathcal{S}_r} \max\{0, \text{score}(h_i, t_i) + \gamma - \text{score}(h_i, t_i')\}, \tag{12}$$

where $\gamma$ is a hyper-parameter that determines the margin to separate positive pairs from negative pairs. $\text{score}(h_i, t_i')$ calculates the score of a negative pair $(h_i, t_i')$ which results from negative sampling of the positive pair $(h_i, t_i) \in \mathcal{S}_r$, i.e. $(h_i, r, t_i') \notin \mathcal{G}$. Till now, we have obtained meta relation representation $\mathcal{R}_{\mathcal{T}_r}$ for each few-shot relation $r$, along with a loss function on the reference set.

## 4.4 MAML BASED TRAINING STRATEGY

Noting that $\mathcal{L}(\mathcal{S}_r)$ in Eq. 12 is task-specific and should be minimized on the target task $\mathcal{T}_r$, we adopt a MAML based training strategy (Finn et al., 2017) to optimize the parameters on each task $\mathcal{T}_r$. The obtained loss on reference set $\mathcal{L}(\mathcal{S}_r)$ is not used to train the whole model but to update intermediate parameters. Please refer to Appendix E for the detailed training scheme of MAML. Specifically, the learned meta representation $\mathcal{R}_{\mathcal{T}_r}$ can be further refined based on the gradient of $\mathcal{L}(\mathcal{S}_r)$:

$$R'_{\mathcal{T}_r} = \mathcal{R}_{\mathcal{T}_r} - l_r \nabla_{\mathcal{R}_{\mathcal{T}_r}} \mathcal{L}(\mathcal{S}_r), \tag{13}$$

where $l_r$ indicates the learning rate. Furthermore, for each target relation $\mathcal{T}_r$, the projection vectors $\mathbf{h}_{pi}$, $\mathbf{r}_{pi}$ and $\mathbf{t}_{pi}$ can also be optimized in the same manner of MAML so that the model can generalize and adapt to a new target relation. Following MAML, the projection vectors are updated as follows:

$$\mathbf{h}'_{pi} = \mathbf{h}_{pi} - l_r \nabla_{\mathbf{h}_{pi}} \mathcal{L}(\mathcal{S}_r), \tag{14}$$

$$\mathbf{r}'_{pi} = \mathbf{r}_{pi} - l_r \nabla_{\mathbf{r}_{pi}} \mathcal{L}(\mathcal{S}_r), \tag{15}$$

$$\mathbf{t}'_{pi} = \mathbf{t}_{pi} - l_r \nabla_{\mathbf{t}_{pi}} \mathcal{L}(\mathcal{S}_r). \tag{16}$$

With the updated parameters, we can project and score each entity pair $(h_j, t_j)$ from the query set $\mathcal{Q}_r$ following the same scheme as reference set and obtain the entity-level loss function $\mathcal{L}(\mathcal{Q}_r)$:

$$\mathbf{h}_{pj} = \mathbf{r}'_{pj}\mathbf{h}'_{pj}{}^{\mathsf{T}}\mathbf{h}_j + \mathbf{I}^{m \times n}\mathbf{h}_j, \tag{17}$$

$$\mathbf{t}_{pj} = \mathbf{r}'_{pj}\mathbf{t}'_{pj}{}^{\mathsf{T}}\mathbf{t}_j + \mathbf{I}^{m \times n}\mathbf{t}_j, \tag{18}$$

$$\text{score}(h_j, t_j) = \| \mathbf{h}_{pj} + R'_{\mathcal{T}_r} - \mathbf{t}_{pj} \|_2, \tag{19}$$

$$\mathcal{L}(\mathcal{Q}_r) = \sum\nolimits_{(h_j, t_j) \in \mathcal{Q}_r} \max\{0, \text{score}(h_j, t_j) + \gamma - \text{score}(h_j, t_j')\}, \tag{20}$$

where $(h_j, t_j')$ is also a negative triplet generated in the same way as $(h_i, t_i')$. The optimization objective for training the whole model is to minimize $\mathcal{L}(\mathcal{Q}_r)$ and $\mathcal{L}_c$ together, given by:

$$\mathcal{L} = \mathcal{L}(\mathcal{Q}_r) + \lambda \mathcal{L}_c, \tag{21}$$

where $\lambda$ is a trade-off hyper-parameter that balances the contributions of $\mathcal{L}(\mathcal{Q}_r)$ and $\mathcal{L}_c$.

## 5 EXPERIMENTS

### 5.1 DATASETS AND EVALUATION METRICS

We conduct experiments on two widely used few-shot KG completion datasets, Nell-One and Wiki-One, which are constructed by (Xiong et al., 2018). For fair comparison, we follow the experimental setup of GMatching (Xiong et al., 2018), where relations associated with more than 50 but less than 500 triplets are chosen for few-shot completion tasks. For each target relation, the candidate entity set provided by GMatching is used. The statistics of both datasets are provided in Table 2. We use 51/5/11 and 133/16/34 tasks for training/validation/testing on Nell-One and Wiki-One, respectively, following the common setting in the literature.

We report both MRR (mean reciprocal rank) and Hits@n ($n = 1, 5, 10$) on both datasets for the evaluation of performance. MRR is the mean reciprocal rank of the correct entities, and Hits@n is the ratio of correct entities that rank

Table 2: Statistics of datasets.

| Dataset | # Relations | # Entities | # Triplets | # Tasks |
|---------|-------------|------------|------------|---------|
| Nell-One | 358 | 68,545 | 181,109 | 67 |
| Wiki-One | 822 | 4,838,244 | 5,859,240 | 183 |

in top $n$. We compare the proposed method against other baseline methods in 1-shot and 5-shot settings, which are the most common settings in the literature.

### 5.2 BASELINES

For evaluation, we compare our proposed method against two groups of state-of-the-art baselines:

**Conventional KG completion methods**: TransE (Bordes et al., 2013), TransH (Wang et al., 2014), DistMult (Yang et al., 2015), ComplEx (Trouillon et al., 2016) and ComplEx-N3 (Lacroix et al., 2018). We use OpenKE (Han et al., 2018) to reproduce the results of these models with hyperparameters reported in the original papers. The models are trained using all triplets from background relations (Xiong et al., 2018) and training relations, as well as relations from all reference sets.

**State-of-the-art few-shot KG completion methods**: GMatching (Xiong et al., 2018), MetaR (Chen et al., 2019), FAAN (Sheng et al., 2020) and FSRL (Zhang et al., 2020). For MetaR (both In-Train and Pre-Train) and FAAN, we directly report results obtained from the original papers. For GMatching, we report the results provided by (Chen et al., 2019) for both 1-shot and 5-shot. As FSRL was initially reported in different settings, where the candidate set is much smaller, we report the results re-implemented by (Sheng et al., 2020) under the same setting with other methods. Due to the fact that the reproduced results of GANA (Niu et al., 2021) is less competitive, we leave GANA out in our comparison. All reported results are produced based on the same experimental setting.

### 5.3 EXPERIMENTAL SETUP

For fair comparison, we use the entity and relation embeddings pretrained by TransE (Bordes et al., 2013) on both datasets, released by GMatching (Xiong et al., 2018), for the initialization of our proposed HiRe. Following the literature, the embedding dimension is set to 100 and 50 for Nell-One and Wiki-One, respectively. On both datasets, we set the number of SAB to 1 and each SAB contains one self-attention head. We apply drop path to avoid overfitting with a drop rate of 0.2. The maximum number of neighbors for a given entity is set to 50, the same as in prior works. For all experiments except for the sensitivity test on the trade-off parameter $\lambda$ in Eq. 21, $\lambda$ is set to 0.05 and the number of false contexts for each reference triplet is set to 1. The margin $\gamma$ in Eq. 12 is set to 1. We apply mini-batch gradient descent to train the model with a batch size of 1,024 for both datasets. Adam optimizer is used with a learning rate of 0.001. We evaluate HiRe on validation set every 1,000 steps and choose the best model within 30,000 steps based on MRR. All models are implemented by PyTorch and trained on 1 Tesla P100 GPU.

### 5.4 COMPARISON WITH STATE-OF-THE-ART METHODS

Table 3 compares HiRe against baselines on Nell-One and Wiki-One under 1-shot and 5-shot settings. In general, conventional KG completion methods are inferior to few-shot KG completion methods, especially udner 1-shot setting. This is expected because conventional KG completion methods are designed for scenarios with sufficient training data. Overall, our HiRe method outperforms all baseline methods under two settings on both datasets, which validates its efficacy for few-shot KG completion. Especially, as the number of reference triplets increases, HiRe achieves

Table 3: Comparison against state-of-the-art methods on Nell-One and Wiki-One. MetaR-I and MetaR-P indicate the In-train and Pre-train of MetaR (Chen et al., 2019), respectively. OOM indicates out of memory.

| Methods | Nell-One | | | | | | | | Wiki-One | | | | | | | |
|---|---|---|---|---|---|---|---|---|---|---|---|---|---|---|---|---|
| | MRR | | Hits@10 | | Hits@5 | | Hits@1 | | MRR | | Hits@10 | | Hits@5 | | Hits@1 | |
| | 1-shot | 5-shot | 1-shot | 5-shot | 1-shot | 5-shot | 1-shot | 5-shot | 1-shot | 5-shot | 1-shot | 5-shot | 1-shot | 5-shot | 1-shot | 5-shot |
| TransE | 0.105 | 0.168 | 0.226 | 0.345 | 0.111 | 0.186 | 0.041 | 0.082 | 0.036 | 0.052 | 0.059 | 0.090 | 0.024 | 0.057 | 0.011 | 0.042 |
| TransH | 0.168 | 0.279 | 0.233 | 0.434 | 0.160 | 0.317 | 0.127 | 0.162 | 0.068 | 0.095 | 0.133 | 0.177 | 0.060 | 0.092 | 0.027 | 0.047 |
| DistMult | 0.165 | 0.214 | 0.285 | 0.319 | 0.174 | 0.246 | 0.106 | 0.140 | 0.046 | 0.077 | 0.087 | 0.134 | 0.034 | 0.078 | 0.014 | 0.035 |
| ComplEx | 0.179 | 0.239 | 0.299 | 0.364 | 0.212 | 0.253 | 0.112 | 0.176 | 0.055 | 0.070 | 0.100 | 0.124 | 0.044 | 0.063 | 0.021 | 0.030 |
| ComplEx-N3 | 0.206 | 0.305 | 0.335 | 0.475 | 0.271 | 0.399 | 0.140 | 0.205 | OOM | OOM | OOM | OOM | OOM | OOM | OOM | OOM |
| GMatching | 0.185 | 0.201 | 0.313 | 0.311 | 0.260 | 0.264 | 0.119 | 0.143 | 0.200 | - | 0.336 | - | 0.272 | - | 0.120 | - |
| MetaR-I | 0.250 | 0.261 | 0.401 | 0.437 | 0.336 | 0.350 | 0.170 | 0.168 | 0.193 | 0.221 | 0.280 | 0.302 | 0.233 | 0.264 | 0.152 | 0.178 |
| MetaR-P | 0.164 | 0.209 | 0.331 | 0.355 | 0.238 | 0.280 | 0.093 | 0.141 | 0.314 | 0.323 | 0.404 | 0.418 | 0.375 | 0.385 | 0.266 | 0.270 |
| FSRL | - | 0.184 | - | 0.272 | - | 0.234 | - | 0.136 | - | 0.158 | - | 0.287 | - | 0.206 | - | 0.097 |
| FAAN | - | 0.279 | - | 0.428 | - | 0.364 | - | 0.200 | - | 0.341 | - | 0.436 | - | 0.395 | - | 0.281 |
| **HiRe** | **0.288** | **0.306** | **0.472** | **0.520** | **0.403** | **0.439** | **0.184** | **0.207** | **0.322** | **0.371** | **0.433** | **0.469** | **0.383** | **0.419** | **0.271** | **0.319** |

Table 4: Ablation study of our proposed HiRe under 3-shot and 5-shot settings on Wiki-One.

| Ablation on ↓ | Components | | | 3-shot | | | | 5-shot | | | |
|---|---|---|---|---|---|---|---|---|---|---|---|
| | MTransD | MRL | Context | MRR | Hits@10 | Hits@5 | Hits@1 | MRR | Hits@10 | Hits@5 | Hits@1 |
| HiRe | ✓ | ✓ | ✓ | 0.355 | 0.467 | 0.412 | 0.298 | 0.371 | 0.469 | 0.419 | 0.319 |
| w/o MTransD-MTransE | ✗ | ✓ | ✓ | 0.340 | 0.433 | 0.391 | 0.288 | 0.342 | 0.454 | 0.408 | 0.289 |
| w/o MTransD-MTransH | ✗ | ✓ | ✓ | 0.342 | 0.456 | 0.415 | 0.272 | 0.347 | 0.457 | 0.411 | 0.281 |
| w/o MAML | ✗ | ✓ | ✓ | 0.255 | 0.375 | 0.331 | 0.190 | 0.286 | 0.388 | 0.334 | 0.238 |
| w/o MRL-AVG | ✓ | ✗ | ✓ | 0.315 | 0.430 | 0.365 | 0.255 | 0.317 | 0.433 | 0.371 | 0.258 |
| w/o MRL-LSTM | ✓ | ✗ | ✓ | 0.314 | 0.409 | 0.354 | 0.266 | 0.320 | 0.436 | 0.385 | 0.261 |
| w/o Context | ✓ | ✓ | ✗ | 0.334 | 0.449 | 0.402 | 0.263 | 0.335 | 0.470 | 0.409 | 0.279 |

larger performance gains because our transformer based MRL can capture more complex triplet-level interactions. This further reinforces HiRe's multi-level relational learning process in return.

As for performance gains in terms of MRR, Hits@10, Hits@5, and Hits@1, HiRe surpasses the second best performer by +3.8%, +7.1%, +6.7%, and +1.4% in 1-shot setting, and by +2.7%, +8.3%, +7.5%, and +0.7% in 5-shot setting on Nell-One. For performance gains on Wiki-One, HiRe outperforms the second best method by +0.8%, +2.9%, +0.8%, and +0.5% in 1-shot setting, and by +3.0%, +3.3%, +2.4%, and +3.8% in 5-shot setting. HiRe achieves large performance improvements in terms of all metrics, proving that leveraging hierarchical relational information enhances the model's generalizability and leads to an overall improvement in performance.

## 5.5 ABLATION STUDY

Our proposed HiRe framework is composed of three key components. To investigate the contributions of each component to the overall performance, we conduct a thorough ablation study on both datasets under 3-shot and 5-shot settings. The detailed results on Wiki-One are reported in Table 4.

**w/o MTransD-MTransE and w/o MTransD-MTransH**: To study the effectiveness of MTransD, we substitute MTransD with TransE and TransH respectively, retaining the MAML based training strategy. Substituting MTransD leads to performance drops at a significant level under both settings, indicating the necessity of constraining entity and relation embeddings while simultaneously considering the diversity of entities and relations.

**w/o MAML**: To demonstrate the efficacy of MAML based training strategy, we remove MAML based training strategy from MTransD and replace it with TransD. In this ablated variant, TransD is applied on the query set after the meta relation representation is learned from the reference set. The significant performance drop suggests that MAML based training strategy is essential for the model to learn generalizable meta knowledge for predicting unseen relations in few-shot settings. This conclusion has also been affirmed by the ablation study in MetaR (Chen et al., 2019).

**w/o MRL-AVG**: To study the impact of transformer based MRL, we replace MRL by simply averaging the embeddings of all reference triplets to generate meta relation representations. This has a profoundly negative effect, resulting in a performance drop of 4% in terms of MRR under 3-shot setting and 5.4% under 5-shot setting. The performance under 3-shot and 5-shot settings are similar, indicating that simplistic averaging fails to take advantage of more training triplets. This validates the importance of MRL to capture triplet-level interactions in learning meta relation representations.

**w/o MRL-LSTM**: To further validate the advantages of leveraging pairwise relational information over sequential information, we replace transformer based MRL with an LSTM to generate meta

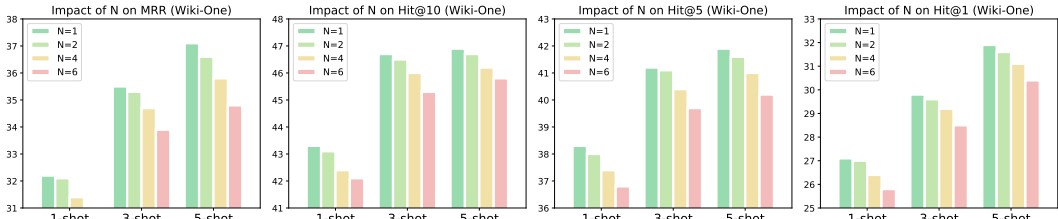

Figure 3: Hyper-parameter sensitivity study with respect to the number of false contexts $N$ on Wiki-One.

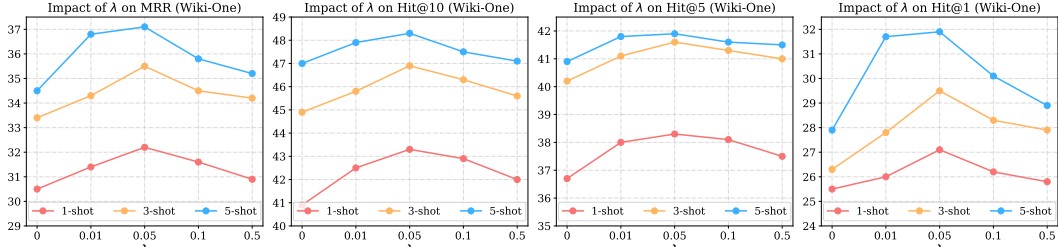

Figure 4: The impact of different $\lambda$ values in Eq. 21 on Wiki-One. $\lambda = 0$ means that we remove the contrastive learning based context-level relational learning.

relation representations. The resultant performance drop is significant; the MRR drops by 4.3% and 5.1% respectively under 3-shot and 5-shot settings. Although the use of LSTM brings some improvements over simplistic averaging through capturing triplet-level relational information, the imposed unrealistic sequential dependency assumption results in limited performance gains. This demonstrates the necessity and superiority of our proposed transformer based MRL in capturing pairwise triplet-level relational information to learn meta representations of few-shot relations.

**w/o Context**: By ablating $\mathcal{L}_c$ from Eq. 21, we remove contrastive learning based context-level relational learning but retain triplet-level and entity-level relational information. As compared to jointly considering the context of the target triplet, the resultant performance drop verifies our assumption that the semantic contextual information plays a crucial role in few-shot KG completion.

Similar conclusions can also be drawn from the ablation results on Nell-One (See Appendix F).

### 5.6 HYPER-PARAMETER SENTITIVITY

We conduct two sensitivity tests for the number of false contexts $N$ and the trade-off parameter $\lambda$ in Eq. 21 on both datasets under $1/3/5$-shot settings. See Appendix G for detailed results on Nell-One.

For hyper-parameter $N$, we set $N$ as 1, 2, 4, and 6. As Figure 3 shows, HiRe performs the best when $N = 1$ on all settings, and its performance slightly drops when $N = 2$. As $N$ continues to increase, the performance of HiRe drops accordingly. One main reason is that, too many false contexts would dominate model training, causing the model to quickly converge to a sub-optimal state.

For hyper-parameter $\lambda$, since the value of contrastive loss is significantly larger than that of the margin loss, $\lambda$ should be small to ensure effective supervision from the margin loss. Thus, we study the impact of different values of $\lambda$ between 0 and 0.5. As Figure 4 shows, HiRe achieves the best performance when $\lambda = 0.05$. With the contrastive loss (*i.e.*, $\lambda > 0$), HiRe consistently yields better performance, proving the efficacy of our contrastive learning based context-level relational learning.

## 6 CONCLUSION

This paper presents a hierarchical relational learning framework (HiRe) for few-shot KG completion. We investigate the limitations of current few-shot KG completion methods and identify that jointly capturing three levels of relational information is crucial for enriching entity and relation embeddings, which ultimately leads to better meta representation learning for the target relation and model generalizability. Experimental results on two commonly used benchmark datasets show that HiRe consistently outperforms current state-of-the-art methods, demonstrating its superiority and efficacy for few-shot KG completion. The ablation analysis and hyper-parameter sensitivity study verify the significance of the key components of HiRe.

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

## APPENDIX

## A NOTATIONS

The notations and symbols used in this paper are summarized in Table 5.

Table 5: Notations and Symbols.

| Symbol | Description |
|---|---|
| $\mathcal{G}$ | knowledge graph |
| $\mathcal{E}, \mathcal{R}, \mathcal{TP}$ | entity, relation and triplet sets of a knowledge graph |
| $h, t$ | head entity, tail entity |
| $r$ | relation |
| $(h, r, t)$ | factual triplet |
| $\mathbf{h}, \mathbf{r}, \mathbf{t}$ | embeddings of $h, r$ and $t$ |
| $\mathcal{T}_r$ | few-shot task corresponding to relation $r$ |
| $\mathcal{S}_r$ | reference set corresponding to relation $r$ |
| $\mathcal{Q}_r$ | query set corresponding to relation $r$ |
| $\mathbb{C}_{(\tilde{h}_j, r)}$ | candidate set for the potential tail entity of $(\tilde{h}_j, r, ?)$ |
| $\mathcal{N}_e$ | set of neighboring relation-entity tuples of entity $e$ |
| $\mathcal{C}_{(h,r,t)}$ | context of triplet $(h, r, t)$ |

## B MOTIVATION: SHARED CONTEXT STATISTICS

One of our key motivations is that jointly considering the wider context shared by head/tail entity would reveal crucial information for learning expressive entity embeddings. To justify our motivation, we perform a statistical analysis on Nell-One and Wiki-One dataset and the results are summarized in Table 6.

Overall, out of the 189,635 triplets in Nell-One dataset, up to 39,234 triplets share entities in their contexts. That means, there exists at least one entity that is connected to both the head entity and the tail entity of the given triplet. These 39,234 triplets share 117,386 entities in all, making each triplet have almost three shared entities by average. Triplets that share entities in their contexts constitute more than 20.68% and 9.2% of the total triplets, respectively. More strictly, the triplets that share relation-entity tuples in their contexts (*i.e.*, meaning that the head and tail entity are connected to the same entity by the same relation in the context) constitute 13.77% on Nell-One and 4.6% on Wiki-One, respectively.

Our analysis affirms that the triplets on KGs indeed share a significant amount of context information. Our method is thus designed to leverage such crucial information for learning more expressive entity embeddings.

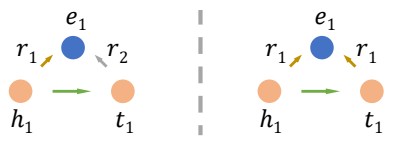

w/ shared entity ($e_1$)    w/ shared tuple ($r_1, e_1$)

(a) Left: triplets that share entity $e_1$; Right: triplets that share relation-entity tuple ($r_1, e_1$).

| | # Tr. | # Tr. w/ shared Ent. | # shared Ent. |
|---|---|---|---|
| Nell-One | 189,635 | 39,234 | 117,368 |
| WiKi-One | 61,498 | 5,665 | 7,753 |

| | # Tr. | # Tr. w/ shared Tup. | # shared Tup. |
|---|---|---|---|
| Nell-One | 189,635 | 26,129 | 105,113 |
| WiKi-One | 61,498 | 2,843 | 3,817 |

(b) Top: triplets (Tr.) that share entities (Ent.) in their contexts; Bottom: triplets (Tr.) that share (relation, entity) tuples (Tup.).

Table 6: Statistical results on Nell-One and WiKi-One. We show the number of triplets that share entities or relation-entity tuples in their contexts. "Shared tuples" means that the head entity and tail entity are connected to the same entity by the same relation in the context, and "shared entities" means that the head entity and tail entity are connected to the same entity by any relation, as illustrated in the left figure. All Numbers are calculated on the training set.

## C  OVERVIEW OF THE PROPOSED HiRE FRAMEWORK

Figure 5 shows an overview of our proposed hierarchical relational learning (HiRe) framework.

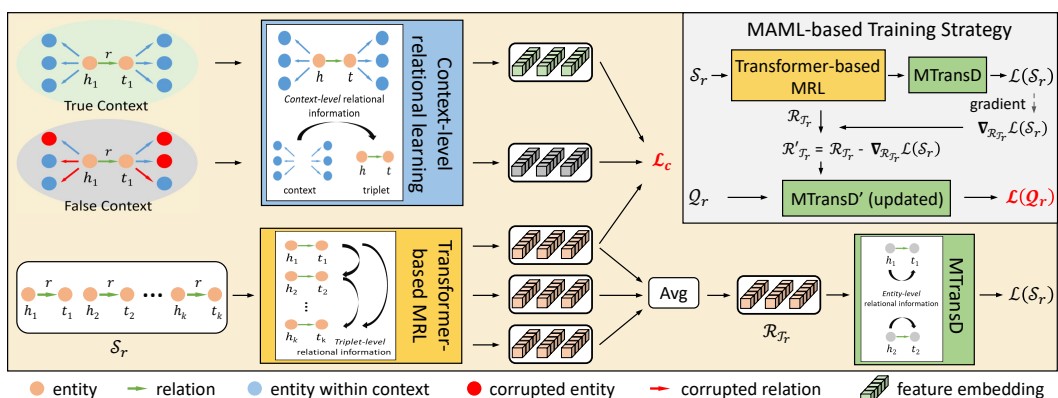

Figure 5: An overview of **HiRe** framework composed of three key components. (1) Contrastive learning based context-level relational learning; (2) Transformer based triplet-level relational learning; (3) Meta representation based entity-level relational learning. Given a target relation $r$ and its corresponding reference set $\mathcal{S}_r$ and query set $\mathcal{Q}_r$, we employ a contrastive loss $\mathcal{L}_c$ between the true/false contexts and the anchor triplet (take $(h_1, r, t_1)$ as an example) via our proposed contrastive learning based context-level relational learning method. The meta representation of the target relation $\mathcal{R}_{\mathcal{T}_r}$ is learned by our Transformer based meta relation learner (MRL), capturing pairwise triplet-level relational information. Lastly, MTransD refines the learned meta relation representation at the entity level constrained by $\mathcal{L}(\mathcal{S}_r)$. The whole learning framework is optimized by a MAML based training strategy.

## D  DETAILS: MULTI-HEAD SELF-ATTENTION AND SET ATTENTION BLOCK

### D.1  DETAILS OF MULTI-HEAT SELF ATTENTION

For context-level relational learning, we employ a Multi-Head Self-Attention (MSA) block (Vaswani et al., 2017) to uncover the underlying relationships within the context $\mathcal{C}_{(h,r,t)}$ of a given triplet $(h, r, t)$ and generate the context embedding $\mathbf{c}$.

Specifically, take a context that contains $k$ relation-entity tuples as an example, the context embedding is initialized as $\mathbf{c} \in \mathbb{R}^{k \times d_c}$ ($d_c = 100$ and $d_c = 200$ for WiKi-One and Nell-One, respectively). This input is first transformed into three different matrices: the query matrix $\mathbf{Q} \in \mathbb{R}^{k \times d_v}$, the key matrix $\mathbf{K} \in \mathbb{R}^{k \times d_k}$ and the value matrix $\mathbf{V} \in \mathbb{R}^{k \times d_v}$ with dimension $d_q = d_k = d_v = d_c$. Subsequently, the attention function is calculated as follows:

- **Step 1**: Compute the scores between different input matrices as $\mathbf{S} = \mathbf{Q} \cdot \mathbf{K}^\top$;
- **Step 2**: Normalize the scores for the stability of gradient as $\mathbf{S}_n = \mathbf{S}/\sqrt{d_k}$;
- **Step 3**: Translate the scores into probabilities with softmax function $\mathbf{A} = \mathrm{softmax}(\mathbf{S}_n)$;
- **Step 4**: Obtain the weighted value matrix $\mathbf{C} = \mathbf{A} \cdot \mathbf{V}$

The above process can be unified into a single function:

$$\mathrm{Attention}(\mathbf{Q}, \mathbf{K}, \mathbf{V}) = \mathrm{softmax}(\frac{\mathbf{Q} \cdot \mathbf{K}^\top}{\sqrt{d_k}}) \cdot \mathbf{V}. \tag{22}$$

The logic behind Eq. 22 is straightforward. Step 1 computes a score between each pair of different relation-entity tuples from the context. Step 2 normalizes the scores to enhance gradient stability for improved training, and Step 3 translates the scores into probabilities. Finally, each relation-entity

tuple is updated by the weighted sum based on the probabilities. Overall, Eq. 3 and Eq. 4 in the main paper can be re-formulated as:

$$\mathbf{c} = \text{Attention}(\mathbf{c}_0) = \text{Attention}(\mathbf{Q}, \mathbf{K}, \mathbf{V}) = \mathbf{A} \cdot \mathbf{V} = \text{softmax}(\frac{\mathbf{Q} \cdot \mathbf{K}^\top}{\sqrt{d_k}}) \cdot \mathbf{V}, \quad (23)$$

where $\mathbf{A} = [\alpha_1; \alpha_2; ...; \alpha_k]$. Instead of performing a single attention function with $d_c$-dimensional queries, keys and values, it would be beneficial to linearly project the queries, keys and values $h$ times with different linear projections (called "multi-head") (Vaswani et al., 2017). On each of these projected versions of queries, keys and values, the attention function is executed in parallel.

$$\text{MultiHead}(\mathbf{Q}, \mathbf{K}, \mathbf{V}) = \text{Concat}(\text{head}_1, ..., \text{head}_h)\mathbf{W}^O, \quad (24)$$

$$\text{where head}_i = \text{Attention}(\mathbf{Q}\mathbf{W}_i^Q, \mathbf{K}\mathbf{W}_i^K, \mathbf{V}\mathbf{W}_i^V), \quad (25)$$

where the projections are parameter matrices $\mathbf{W}_i^Q \in \mathbb{R}^{d \times d_k}$, $\mathbf{W}_i^K \in \mathbb{R}^{d \times d_k}$, $\mathbf{W}_i^V \in \mathbb{R}^{d \times d_k}$ and $\mathbf{W}^O \in \mathbb{R}^{hd_v \times d}$. In each paralleled attention layer, $d_k = d_v = d/h$.

## D.2 Details of Set Attention Block

To comprehensively incorporate triplet-level relational information in the reference set $\mathcal{S}_r$, we design a transformer based MRL using a set attention block (SAB) (Lee et al., 2019) to model interactions among all reference triplets in $\mathcal{S}_r$.

SAB is built upon multi-head attention, defined as:

$$\text{SAB}(X) := \text{LayerNorm}(H + \text{rFF}(H)) \quad (26)$$

where $H = \text{LayerNorm}(X + \text{MultiHead}(X, X, X))$, rFF is any row-wise feedforward layer and LayerNorm is layer normalization (Ba et al., 2016).

## E MAML based Training Strategy

The detailed MAML based training framework can be described as follows:

---

**Algorithm 1:** MAML based training framework of HiRe.

**Input:** $\mathcal{T}_{train}$: Training tasks
      $\mathcal{G}_b$: Background graph

1 **while** *not converged* **do**
2    Sample a task $\mathcal{T}_r = \{\mathcal{S}_r, \mathcal{Q}_r\}$ from $\mathcal{T}_{train}$;
3    Construct context $\mathcal{C}_{(h,r,t)}$ for each reference triplet $(h, r, t)$ in $\mathcal{S}_r$ based on $\mathcal{G}_b$;
4    Encode $\mathcal{C}_{(h,r,t)}$ and produce context embedding $\mathbf{c}$ by Eq. 3-Eq. 4;
5    Learn context-level relational information based on contrastive learning by Eq. 5;
6    Learn triplet-level meta representation for relation $r$ via transformer based MRL by Eq. 6-Eq. 8;
7    Learn entity-level relational information via MTransD by Eq. 9-Eq. 11;
8    Calculate the loss on reference set $\mathcal{L}(\mathcal{S}_r)$ by Eq. 12;
9    Update the parameters based on $\mathcal{L}(\mathcal{S}_r)$ by Eq. 13-Eq. 16;
10    Calculate the loss on query set $\mathcal{L}(\mathcal{Q}_r)$ by Eq. 17-Eq. 20;
11    Update the model parameters based on the overall loss function Eq. 21

---

## F Ablation Study on Nell-One

As discussed in Section 5.5, each component in our proposed HiRe framework plays an important role in few-shot KG completion. Here, we provide further ablation results on Nell-One in Table 7. These results support our findings reported in the main paper and confirm that the removal of any component leads to performance drops in terms of all evaluation metrics.

Table 7: Ablation study of HiRe under 3-shot and 5-shot settings on Nell-One.

| Ablation on ↓ | Components | | | 3-shot | | | | 5-shot | | | |
|---|---|---|---|---|---|---|---|---|---|---|---|
| | MTransD | MRL | Context | MRR | Hits@10 | Hits@5 | Hits@1 | MRR | Hits@10 | Hits@5 | Hits@1 |
| HiRe | ✓ | ✓ | ✓ | 0.300 | 0.499 | 0.425 | 0.199 | 0.306 | 0.520 | 0.439 | 0.207 |
| w/o MTransD-MTransE | ✗ | ✓ | ✓ | 0.295 | 0.491 | 0.420 | 0.193 | 0.302 | 0.515 | 0.428 | 0.202 |
| w/o MTransD-MTransH | ✗ | ✓ | ✓ | 0.293 | 0.494 | 0.418 | 0.191 | 0.303 | 0.513 | 0.430 | 0.203 |
| w/o MAML | ✗ | ✓ | ✓ | 0.177 | 0.305 | 0.251 | 0.105 | 0.198 | 0.334 | 0.271 | 0.123 |
| w/o MRL-AVG | ✓ | ✗ | ✓ | 0.282 | 0.467 | 0.382 | 0.185 | 0.286 | 0.466 | 0.394 | 0.188 |
| w/o MRL-LSTM | ✓ | ✗ | ✓ | 0.280 | 0.490 | 0.410 | 0.168 | 0.285 | 0.459 | 0.390 | 0.185 |
| w/o Context | ✓ | ✓ | ✗ | 0.290 | 0.482 | 0.401 | 0.186 | 0.295 | 0.489 | 0.418 | 0.197 |

## G HYPER-PARAMETER SENSITIVITY STUDY ON NELL-ONE

Figure 6 and Figure 7 report further sensitivity test results on Nell-One for the number of false contexts $N$ and the trade-off parameter $\lambda$. As shown in Figure 6, as the number of false contexts increases, the performance of HiRe drops slightly because its model training would converge to a sub-optimal state. Moreover, the best $\lambda$ value is also $0.05$ on Nell-One, as shown in Figure 7. The overall findings are consistent with those we draw from the results on Wiki-One in Section 5.6.

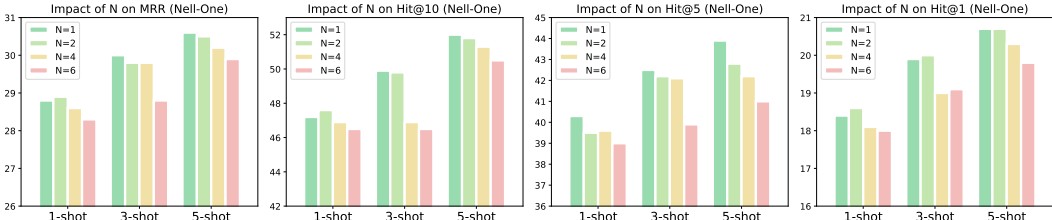

Figure 6: Hyper-parameter sensitivity study with respect to the number of false contexts on Nell-One.

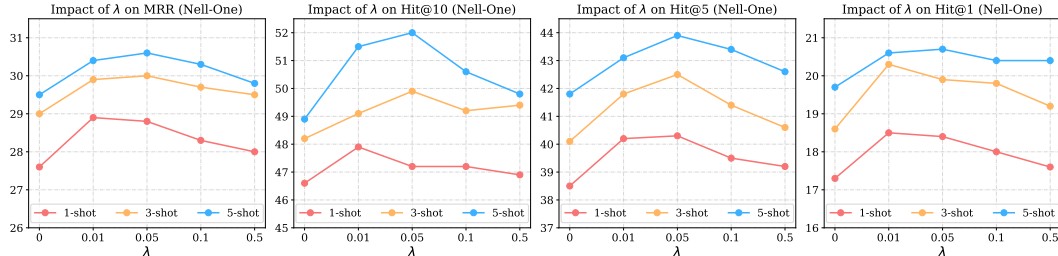

Figure 7: The impact of different $\lambda$ values in Eq. 21 on Nell-One. $\lambda = 0$ means that we remove the contrastive learning based context-level relational learning.

## H COMPLEXITY ANALYSIS

Table 8 lists the complexity of all the hierarchical relational learning modules, where $d$ denotes the dimension of entity embeddings, $k$ denotes the number of relation-entity tuples in the context, and $n$ denotes the number of reference triplets in each task. As can be seen, our proposed HiRe quadratically scales with the number of reference triplets in each task. Nevertheless, given the number of reference triplets is often very small (*i.e.*, 1, 3, 5) , our proposed HiRe framework scales reasonably well.

Table 8: Complexity analysis of three hierarchical relational learning modules with respect to the number of parameters and the number of multiplication operations in each epoch.

| | # Parameters | # Operations |
|---|---|---|
| Context level | $\mathcal{O}(nkd + nk^2)$ | $\mathcal{O}(nkd^2 + nk^2d)$ |
| Triplet level | $\mathcal{O}(nd + n^2)$ | $\mathcal{O}(nd^2 + n^2d)$ |
| Entity level | $\mathcal{O}(d)$ | $\mathcal{O}(d)$ |
| Total | $\mathcal{O}(nkd + nk^2 + n^2)$ | $\mathcal{O}(nkd^2 + nk^2d + n^2d)$ |

