# OpenReview forum: "Hierarchical Relational Learning for Few-Shot Knowledge Graph Completion"
_ICLR.cc/2023/Conference — ICLR 2023 poster_

### Official Review · Reviewer_8oAh · 2022-10-23

**Confidence:** 5
**Correctness:** 2
**Technical Novelty And Significance:** 2
**Empirical Novelty And Significance:** 2
**Recommendation:** 5

**Clarity, Quality, Novelty And Reproducibility:**

Clarity:
1. The claim of “the potential of leveraging context-level relation information” is inaccurate because the existing work aggregates information from the local neighbors.
2. The authors claim the existing works lose critical information. First, I wonder how many triples will have shared information. Second, why the previous methods cannot capture the shared entity? I think the previous methods implicitly encode the shared entity in their embeddings. The claim needs more explanations.
3. The motivation for using the transformer-based meta-relation learner is not clear. Why is the utilization of LSTM based on an unrealistic assumption and what is the impact caused by the insensitive of the size of the reference set and the permutation-invariant? And Why these two properties are essential?
4. The advantages of the proposed MTransD are not clarified. Why not use the existing translation-based methods? And what is the disadvantage of these methods?
5. The ablation study is necessary to demonstrate the advantage of MTransD compared with TransE, RotateE… the related algorithms.

Quality and Novelty:
This work replaces the key components in existing works and proposes an incremental solution to improve the performance of few-shot KG completion. The proposed method is reasonable but lacks novelty.

Reproducibility:
The authors provide the data and code for reproducing their results.

**Strength And Weaknesses:**

Strengths:
1. The paper is well-organized and the proposed method is easy to follow.
2. The studied problem is an important research problem for KG completion.

Weaknesses:
1. The proposed method is based on previous works and uses different components to replace the existing ones. The incremental improvement shows the lack of novelty in this work.
2. The motivation for the proposed method is not clear and hard to understand why the proposed design is necessary.
3. The lack of an ablation study to demonstrate the superiority of the MTransD.


**Summary Of The Paper:**

The paper studies the few-shot KG completion problem and proposes a hierarchical relation learning method for this task. The authors propose three levels of relational information to learn and refine the meta-representation of few-shot relations. Specifically, a contrastive learning-based context-level relation learning is proposed to update the entity embeddings, and a transformer-based triplet-level relation learning is proposed to learn the meta-relation representation, and an MTransD is proposed as the score function. The authors conduct experiments on widely used datasets.

**Summary Of The Review:**

The paper proposes an incremental solution for the few-shot KG completion and lacks novelty to some extent. And some claims and motivations are not clear and need further explanations. Thus I think the quality of this paper is below the standard bar of ICLR and would not like to accept it.

---

> ### Author Response · Authors · 2022-11-15
> **Response to Reviewer 8oAh (Part 1/4)**
>
> We sincerely thank you for your valuable comments. We respond to your comments and provide clarifications below.
>
> >**Question 1:** Lack of novelty. The proposed method is based on previous works and uses different components to replace the existing ones.
>
> **Response to question 1:**  We hope to explain the philosophy behind the principles that underpin our research work. By examining previous works and understanding the possible weaknesses therein, we have been able to correctly identify valid research gaps/issues in the existing few-shot KG completion techniques. These strongly motivate a compelling need for novel methodology to address these gaps. Second, we propose a new solution which is specifically targeted to address these important research gaps. In particular, we have carefully designed and employed appropriate components. Third, our experimental evaluation demonstrates the superiority of our proposed solution, affirming the novelty of our contributions.
>
> (1) As mentioned in the paper, we reiterate the following important research gaps, which motivate the need for developing novel solutions:
>
> - Research gap 1: the existing works have imposed an unrealistic sequential assumption in the reference set to learn meta relation information.
> - Research gap 2: The potential of leveraging pairwise triplet-level interactions and context-level relational information has been largely unexplored.
>
> (2) The novel aspects of our method are as follows:
>
> - To our best knowledge, our method is the first to model pairwise interactions among reference triplets towards learning meta relation representation, while preserving the desirable permutation-invariant property.
> - We are the first to propose a contrastive learning based method to learn expressive entity/relation embeddings by modeling correlations between the target triplet and its true/false contexts.
>
> (3) Very importantly, the novelty of our solution leads to new state-of-the-art results, outperforming the existing methods on widely used benchmark datasets.
>
> In addition to the above verifiable facts, the novelty of our work has also been endorsed by **Reviewer Pexb** and **Reviewer b64W** as follows:
>
> Pexb: "The paper identifies valid gaps/issues in previous techniques and provides sensical first steps to address them."
>
> b64W: "The authors very succinctly point out the gap in the existing few-shot paradigm, viz, inconsistent sequential assumption, and lack of context awareness."
>
> b64W: "The paper correctly identifies the problem with the existing approaches and addresses them with noble techniques."
>
> > **Question 2:** The motivation for the proposed method is not clear and hard to understand why the proposed design is necessary.
>
> **Response to question 2:** Thanks for the comment. As we discussed in our *response to limitation 1*, the motivation of the proposed method is based on the two identified research gaps. We will discuss in more details in *response to question 4/5/6*.
>
> > **Question 3:** The lack of an ablation study to demonstrate the superiority of the MTransD.
>
> **Response to question 3:** Thanks for this question. We would like to bring forward two important points.
>
> First, we had already demonstrated the superiority of MTransD by ablating MTransD with MTransE (while retaining the MAML-based training strategy) in our ablation study. In order to provide further insights, we have also added a new experiment of ablating MTransD with MTransH. The results in Table 4 demonstrated that substituting TransD with other translation-based methods (while retaining MAML) leads to significant performance drops.
>
> Second, in order to respond adequately to the reviewer's comments, we have added a new ablation study to demonstrate the efficacy of MAML-based training strategy as part of MTransD. This is undertaken by removing MAML-based training strategy from MTransD and replacing it with the tranditional TransD. In this ablated variant, TransD is applied on the query set after the meta relation representation is learned from the reference set.
>
> The new ablation results on Nell-One under 3-shot and 5-shot setting are as follows.
>
> |     3-shot     |  MRR  | Hits@10 | Hits@5 | Hits@1 |
> | :------------: | :---: | :-----: | :----: | :----: |
> |    w/o MAML    | 0.177 |  0.305  | 0.251  | 0.105  |
> | w/ MAML (HiRe) | 0.300 |  0.499  | 0.425  | 0.199  |
>
> |     5-shot     |  MRR  | Hits@10 | Hits@5 | Hits@1 |
> | :------------: | :---: | :-----: | :----: | :----: |
> |    w/o MAML    | 0.198 |  0.334  | 0.271  | 0.123  |
> | w/ MAML (HiRe) | 0.306 |  0.520  | 0.439  | 0.207  |
>
> The results suggest that MAML-based training strategy is essential for the model to learn generalizable meta knowledge to predict unseen relations in few-shot settings. This conclusion has also been affirmed by the ablation study of MetaR [13] (see Table 5 in the original paper).
>
> The above new ablation results have been added to Table 7 in Appendix F and Table 5 in the revised paper.

---

> > ### Author Response · Authors · 2022-11-15
> > **Response to Reviewer 8oAh (Part 2/4)**
> >
> > > **Question 4:** The claim of “the potential of leveraging context-level relation information” is inaccurate because the existing work aggregates information from the local neighbors.
> >
> > **Response to question 4:** We need to point out a potential misunderstanding here. In our work, "context-level relational information" refers to the correlations between the target triplet and its true context, where the true context is the union of head and tail entity neighborhoods. Naturally,  one triplet itself should have a close relationship with its true context. Thus, we take a contrastive approach --- a given triplet should be pulled close to its true context, but pushed apart from its false contexts --- to learn expressive entity embeddings.
> >
> > However, all the existing works simply regard local neighbors of head/tail entity as its context separately. As a result, the crucial information contained in the wider context is overlooked.
> >
> > We have provided further clarification in Section 4.1 as follows:
> >
> > "Formally, given a target triplet $(h, r, t)$, we denote its wider context as $\mathcal{C}_{(h,r,t)} = \mathcal{N}_h\cup\mathcal{N}_t$, where $\mathcal{N}_h=\{(r_i, t_i)|(h, r_i, t_i) \in \mathcal{TP}\}$ and $\mathcal{N}_t=\{(r_j, t_j)|(t, r_j, t_j) \in \mathcal{TP}\}$. Our goal is to capture context-level relational information --- the correlation between the given triplet and its true context."
> >
> > > **Question 5:** The authors claim the existing works lose critical information. First, I wonder how many triples will have shared information. Second, why the previous methods cannot capture the shared entity? I think the previous methods implicitly encode the shared entity in their embeddings. The claim needs more explanations.
> >
> > **Response to question 5:** Thanks for your questions. We have carefully addressed your questions and provided more explanations below.
> >
> > (1). First of all, we have carried out a new analysis to obtain statistics regarding the number of triplets that have shared information. Please find an illustration of shared information using the following anonymous link: https://i.postimg.cc/hvhK7by9/shared-entity-pair.png
> >
> > The statistical results are summarized in the tables below.
> >
> > Overall, out of the 189,635 triplets in Nell-One dataset, up to 39,234 triplets have shared entities in their contexts. That means, there exists at least one entity that is connected to both the head entity and the tail entity of the given triplet. These 39,234 triplets share 117,386 entities in all, making each triplet have almost three shared entities by average. Triplets that have shared entities in their contexts constitute more than 20.68\% and 9.2\% of the total triplets, respectively. More strictly, the triplets that have shared (relation, entity) pairs in their contexts (meaning the head and tail entity are connected to the same entity by the same relation in the context) constitute 13.77\% on Nell-One and 4.6\% on Wiki-One, respectively.
> >
> > Our statistical analysis reveals that the triplets in KGs indeed share a significant amount of context information. Our method is thus designed to leverage such crucial information for learning more expressive entity embeddings. The importance of such context information has also been validated through our ablation study in Section 5.5.
> >
> > |          | Number of triplets | Number of triplets \w shared entity | Total number of shared entities |
> > | -------- | :----------------: | :---------------------------------: | :-----------------------------: |
> > | Nell-One |       189635       |                39234                |             117368              |
> > | Wiki-One |       61498        |                5665                 |              7753               |
> >
> > |          | Number of triplets | Number of triplets \w shared (relation, entity) | Total number of shared pairs |
> > | -------- | :----------------: | :---------------------------------------------: | :--------------------------: |
> > | Nell-One |       189635       |                      26129                      |            105113            |
> > | Wiki-One |       61498        |                      2843                       |             3817             |
> >
> > We have added this analysis and related discussions in Appendix B in the updated version to provide justification for our motivation.
> >
> > (2). We believe there is an unfortunate misunderstanding that we clarify below. Previous works simply aggregate neighbors of the head/tail entity separately and are thus unaware of any entities/relations shared by both the head and tail entity. In contrast, our work jointly considers the wider context shared by the head and tail entity, which includes the shared entities and shared relations.

---

> > > ### Author Response · Authors · 2022-11-15
> > > **Response to Reviewer 8oAh (Part 3/4)**
> > >
> > > > **Question 6:** The motivation for using the transformer-based meta-relation learner is not clear. Why is the utilization of LSTM based on an unrealistic assumption and what is the impact caused by the insensitive of the size of the reference set and the permutation-invariant? And Why these two properties are essential?
> > >
> > > **Response to question 6:** Thanks for your comments. We aim to systematically clarify the multiple questions below:
> > >
> > > (1) We believe that previous works utilize LSTMs to learn meta representation of a given relation, which inevitably imposes an unrealistic sequential dependence assumption on the reference triplets, because LSTMs are designed to model sequence data. In fact, triplets in the reference set are associated with a same relation and are not sequentially dependent on each other. That means, the occurrence of one triplet does not necessarily lead to other triplets in the reference set.
> > >
> > > (2) Given a reference set, the main focus is to learn the commonality of the reference triplets, which do not have a natural order. The ordering imposed by LSTMs is not present within the reference set. Thus, imposing an order that does not exist can lead to instability. Changing the order of these triplets or the size of reference set will both dramatically impact the learned meta representation.
> > >
> > > (3) The third problem of using LSTMs is information loss, as model training heavily depends on the information carried from the preceding triplets. On the contrary, in our transformer-based MRL, each triplet has **direct** access to all the other triplets, thus eliminating information loss during model training. Considering the scarcity of training data in few-shot settings, making the use of available information is especially critical. The advantage of our transformer-based MRL over the LSTM-based approaches is that, our transformer-based MRL is agnostic to the size of reference set and can captures pairwise interaction among the reference triplets to learn generalizable meta representation of few-shot relations.
> > >
> > > In a broader sense, retaining permutation-invariance is a common practice in the field of graph neural networks. A similar case is designing neighbor aggregators which learn to aggregate the information in the neighborhood. The neighbors, like the reference triplets in our case, have no sequential dependency with each other. Another popular consensus is that in graph classification tasks, the pooling layer should be invariant to the node permutation since the nodes are not sequentially dependent on each other. There are multiple works in the literature that affirm the importance of this property [8-12].
> > >
> > > We have added the following sentences to Section 4.2 to provide more explanations:
> > >
> > > "...which inevitably imposes an unrealistic sequential dependency assumption on reference triplets since LSTMs are designed to model sequence data. However, these triplets are associated with the same relation and are not sequentially dependent on each other; the occurrence of one reference triplet does not necessarily lead to other triplets in the reference set."

---

> > > > ### Author Response · Authors · 2022-11-15
> > > > **Response to Reviewer 8oAh (Part 4/4)**
> > > >
> > > > > **Question 7:** The advantages of the proposed MTransD are not clarified. Why not use the existing translation-based methods? And what is the disadvantage of these methods?
> > > >
> > > > **Response to question 7:** Thank you for the comment. We will reiterate the multiple advantages of the proposed MTransD below:
> > > >
> > > > Overall, the design of MTransD is to warrant the generalizability of the learned meta relation representation at the entity level. Its MAML-based training strategy enables the meta relation representation learned on the reference set to quickly adapt to the query set. The learning of meta representation on both sets is optimized using TransD as the constraint over true/false triplets, since the meta relation representation should hold between true entity pairs (and not hold between false entity pairs).
> > > >
> > > > The existing translation-based methods are designed for data sufficient scenarios and cannot deal with unseen relations that have never appeared during training. But in few-shot KG completion, each relation is only associated with very few training triplets and all relations in the testing stage are novel. That means, directly applying existing translation-based methods to few-shot KG completion would inhibit model from achieving generalizability. As indicated in our response to question 3, directly using TransD without MAML-based training strategy leads to a significant performance drop.
> > > >
> > > > **Reference:**
> > > >
> > > > [8] Ying, Zhitao, et al. "Hierarchical graph representation learning with differentiable pooling." *Advances in neural information processing systems* 31 (2018).
> > > >
> > > > [9] Soheil Kolouri, Navid Naderializadeh, Gustavo K. Rohde, and Heiko Hoffmann. Wasserstein embedding for graph learning. In International Conference on Learning Representations, 2021.
> > > >
> > > > [10] Keyulu Xu, Weihua Hu, Jure Leskovec, and Stefanie Jegelka. How powerful are graph neural networks? In International Conference on Learning Representations, 2019.
> > > >
> > > > [11] Muhan Zhang, Zhicheng Cui, Marion Neumann, and Yixin Chen. An end-to-end deep learning architecture for graph classification. In AAAI Conference on Artificial Intelligence, pp. 4438–4445, 2018.
> > > >
> > > > [12] Winter, Robin, Frank Noé, and Djork-Arné Clevert. "Permutation-invariant variational autoencoder for graph-level representation learning." *Advances in Neural Information Processing Systems* 34 (2021): 9559-9573.
> > > >
> > > > [13] M. Chen, W. Zhang, W. Zhang, Q. Chen, and H. Chen. 2019. Meta relational learning for few-shot link prediction in knowledge graphs. In Proceedings of the 2019 Conference on Empirical Methods in Natural Language Processing. 4217–4226.

---

> ### Author Response · Authors · 2022-12-07
> **Thanks for the comments.**
>
> Dear Reviewer 8oAh,
>
> Thanks for your constructive review. Has our response resolved your concerns? If there are other questions, we are glad to discuss them with you.
>
> Regards

---

### Official Review · Reviewer_HQvj · 2022-10-24

**Confidence:** 4
**Clarity, Quality, Novelty And Reproducibility:** 1. The novelty of this work is limite…
**Correctness:** 3
**Technical Novelty And Significance:** 2
**Empirical Novelty And Significance:** 3
**Recommendation:** 5

**Strength And Weaknesses:**

Strength:

1. The idea of using multiple different levels of information for relational learning on KGs is interesting;

2. The paper is well-written and the technical details are basically presented clearly;

3. The experimental results demonstrate significant improvements over baseline methods.

Weaknesses:

1. The novelty of this work is limited;

2. The term "hierarchical" in the title is confusing;

3. Some of the technical details are not clearly presented;

See below for details of weaknesses.

**Summary Of The Paper:**

This paper proposes a hierarchical relational learning framework for few-shot KG completion. The authors proposes jointly capturing three levels of relational information for enriching entity and relation embeddings, i.e., entity-level, triplet-level and context-level. Experiments on two benchmark datasets demonstrate the effectiveness of the proposed model.

**Summary Of The Review:**

This paper studied an interesting problem in KG representation learning. The paper is basically well-written and the experimental results are promising. However, the technical novelty is limited, and some of the technical details are not clearly presented. Overall, I think this paper is marginally below the acceptance threshold of ICLR.

---

> ### Author Response · Authors · 2022-11-15
> **Response to Reviewer HQvj (Part 1/2)**
>
> We sincerely thank for your valuable comments. We have updated the rebuttal revision accordingly. We respond to your comments and provide clarifications as follows.
>
> > **Question 1:** The novelty of this work is limited.
>
> **Response to question 1:**  We hope to explain the philosophy behind the principles that underpin our research work. By examining previous works and understanding the possible weaknesses therein, we have been able to correctly identify valid research gaps/issues in the existing few-shot KG completion techniques. These strongly motivate a compelling need for novel methodology to address these gaps. Second, we propose a new solution which is specifically targeted to address these important research gaps. In particular, we have carefully designed and employed appropriate components. Third, our experimental evaluation demonstrates the superiority of our proposed solution, affirming the novelty of our contributions.
>
> (1) As mentioned in the paper, we reiterate the following important research gaps, which motivate the need for developing novel solutions:
>
> - Research gap 1: the existing works have imposed an unrealistic sequential assumption in the reference set to learn meta relation information.
> - Research gap 2: The potential of leveraging pairwise triplet-level interactions and context-level relational information has been largely unexplored.
>
> (2) The novel aspects of our method are as follows:
> - To our best knowledge, our method is the first to model pairwise interactions among reference triplets towards learning meta relation representation, while preserving the desirable permutation-invariant property.
> - We are the first to propose a contrastive learning based method to learn expressive entity/relation embeddings by modeling correlations between the target triplet and its true/false contexts.
>
> (3) Very importantly, the novelty of our solution leads to new state-of-the-art results, outperforming the existing methods on widely used benchmark datasets.
>
> In addition to the above verifiable facts, the novelty of our work has also been endorsed by **Reviewer Pexb** and **Reviewer b64W** as follows:
>
> Pexb: "The paper identifies valid gaps/issues in previous techniques and provides sensical first steps to address them."
>
> b64W: "The authors very succinctly point out the gap in the existing few-shot paradigm, viz, inconsistent sequential assumption, and lack of context awareness."
>
> b64W: "The paper correctly identifies the problem with the existing approaches and addresses them with noble techniques."
>
> > **Question 2:** The term "hierarchical" in the title is confusing.
>
> **Response to question 2:** Thanks for the comment. The term "hierarchical" in our work refers to relational learning on knowledge graphs performed at three different levels of granularity. We have clarified this in the updated version of the paper and the following has been added.
>
> "In this paper, we propose a Hierarchical Relational learning framework (HiRe) for few-shot KG completion. HiRe jointly models three levels of relational information (entity-level, triplet-level, and context-level) within each few-shot task as mutually reinforcing sources of information to generalize to few-shot relations. Here, "hierarchical" references relational learning performed at three different levels of granularity."
>
> > **Question 3:** When using contrastive learning to construct a false context for a given triplet, are all (r, t) pairs in context corrupted, or only one (r, t) pair is corrupted?
>
> **Response to question 3:** Thanks for raising this important point. In our current design, all (r, t) pairs in the context are randomly corrupted. In this paper, we mainly focus on the design of contrastive pairs in knowledge graphs and how the number of negative contexts affects model performance (please refer to Section 5.6 in the main paper). We believe it is an important issue to study different strategies for generating false contexts and we intend to explore this area further in future work.
>
> > **Question 4:** For the two parts of the loss, i.e., $Q_r$ and $L_c$, are entity and relation embeddings shared across the two modules, or they have two sets of separate embeddings?
>
> **Response to question 4:** The entity embeddings are shared across the two modules. The contrastive loss does not involve meta representation of the relation. We have added more technical details in the updated version. The following has been added to  Section 4.2 in the revised paper.
>
> "Note that the same entity embeddings $\textbf{h}_i$ and $\textbf{t}_i$ are used here as in Eq. 5"

---

> > ### Author Response · Authors · 2022-11-15
> > **Response to Reviewer HQvj (Part 2/2)**
> >
> > > **Question 5:** For the MTransD part, did the authors try other translational-based methods, such as TransE, TransR, TransH?
> >
> > **Response to question 5:** Yes, we have already reported the results of ablating MTransD with TransE (see Table 4 in the main paper). For more comparison, we have also conducted a new ablation experiment of using TransH in place of TransD. We had also considered using TransR, but have omitted it here as TransR has a high complexity and is shown to perform worse than TransD [6].
> >
> > The ablation results of 3-shot and 5-shot setting on Nell-One are as follows.
> >
> > |       3-shot        |  MRR  | Hits@10 | Hits@5 | Hits@1 |
> > | :-----------------: | :---: | :-----: | :----: | :----: |
> > | w/o MTransD-MTransE | 0.295 |  0.491  | 0.420  | 0.193  |
> > | w/o MTransD-MTransH | 0.293 |  0.494  | 0.418  | 0.191  |
> > |   MTransD (HiRe)    | 0.300 |  0.499  | 0.425  | 0.199  |
> >
> > |       5-shot        |  MRR  | Hits@10 | Hits@5 | Hits@1 |
> > | :-----------------: | :---: | :-----: | :----: | :----: |
> > | w/o MTransD-MTransE | 0.302 |  0.515  | 0.428  | 0.202  |
> > | w/o MTransD-MTransH | 0.303 |  0.513  | 0.430  | 0.203  |
> > |   MTransD (HiRe)    | 0.306 |  0.520  | 0.439  | 0.207  |
> >
> > We have added the new results in the revised paper. The corresponding results on Wiki-One and Nell-One can be found in Table 4 in the revised main paper and in Table 7 in the Appendix F, respectively.
> >
> > > **Question 6:** For the context, the authors only consider one-hop neighbors as the context information, but prior work has shown that more neighbors are helpful to model entities and triplets. It is recommended that the authors try to increase the hop of neighbors to see if the model performance can be further improved.
> >
> > **Response to question 6:** Thanks for your suggestion. We have added an experiment that constructs two-hop neighbors as the context. The results under 5-shot setting on Nell-One are as follows:
> > |                        |  MRR  | Hits@10 | Hits@5 | Hits@1 |
> > | :--------------------: | :---: | :-----: | :----: | :----: |
> > |    two-hop neighor     | 0.293 |  0.489  | 0.412  | 0.192  |
> > | one-hop neghbor (HiRe) | 0.306 |  0.520  | 0.439  | 0.207  |
> >
> > From the above table, we can observe that the use of two-hop neighbors as the context leads to a performance drop, as compared with using one-hop neighbors. This might be due to the fact that many entities have only few one-hop neighbors that are closely related to them [7]. Considering two-hop neighbors as the context might significantly introduce noisy information, resulting in performance drops.
> >
> > **Reference:**
> >
> > [6] Guoliang Ji, Shizhu He, Liheng Xu, Kang Liu, and Jun Zhao. Knowledge graph embedding via dynamic mapping matrix. In ACL, pp. 687–696, 2015.
> >
> > [7] Guanglin Niu, Yang Li, Chengguang Tang, Ruiying Geng, Jian Dai, Qiao Liu, Hao Wang, Jian Sun, Fei Huang, and Luo Si. Relational learning with gated and attentive neighbor aggregator for few-shot knowledge graph completion. In SIGIR, pp. 213–222, 2021.

---

> ### Author Response · Authors · 2022-12-07
> **Thanks for the comments.**
>
> Dear Reviewer HQvj,
>
> Thanks for your constructive review. Has our response resolved your concerns? If there are other questions, we are glad to discuss them with you.
>
> Regards

---

### Official Review · Reviewer_b64W · 2022-10-27

**Confidence:** 3
**Clarity, Quality, Novelty And Reproducibility:** N/A, see weakness.
**Correctness:** 4
**Technical Novelty And Significance:** 3
**Empirical Novelty And Significance:** 3
**Recommendation:** 8

**Strength And Weaknesses:**

Strength:
1. The authors very succinctly point out the gap in the existing few-shot paradigm, viz, inconsistent sequential assumption, and lack of context awareness.
2. Their extensive ablation study indicates the necessity of each component of the model, e.g., the set-based transformer used in triplet-level relation capture was justified by introducing simple sum and LSTMs.

Weakness:
 A few places where complicated model choices were not motivated properly. For example, why authors picked the TransE or distance-based framework for entity-level generalization? The pairwise interaction of the triples could have been done using any sort of permutation invariant operation, why SAB was picked?

**Summary Of The Paper:**

- Real-world KGs are incomplete and suffer from the long-tail distribution over the relations.
- Thus the performance (on completion tasks) on the low-frequency relations is poor.
- Prediction of the tail entity given the (head, relation, ?), is considered a few-shot completion problem, in which one could try to learn meta representation for these relations from the limited amount of reference (head, relation, tail) observed.
- In order to learn the meta-relation representation, the authors try to jointly capture three levels of relational information:
  a. Context level b. Triplet level c. Entity level.
- Finally, they propose a meta-learning-based optimization approach to obtain an effective embedding for the tail relation that achieves a few-shot generalization

**Summary Of The Review:**

The proper correctly identifies the problem with the existing approaches and addresses them with noble techniques. For that reason I feel the paper is a suitable candidate for acceptance.

---

> ### Author Response · Authors · 2022-11-15
> **Response to Reviewer b64W**
>
> We sincerely thank the Reviewer for providing supportive comments and acknowledging the technical soundness and novelty of our work. We respond to your comments and provide clarifications as follows.
>
> > **Question 1:** A few places where complicated model choices were not motivated properly. For example, why authors picked the TransE or distance-based framework for entity-level generalization?
>
> **Response to the question 1:** Thank you for pointing this out. We have provided more explanations in the updated paper.
>
> For entity-level generalization, the meta representation of a given relation should hold between true entity pairs (but not hold between false entity pairs). This requires learning relation-specific meta knowledge from a set of reference triplets. Distance-based frameworks provide an intuitive solution by using the relation as a translation, enabling to explicitly model and constrain the learning of generalizable meta knowledge at the entity level.
>
> Furthermore, among all distance-based models, TransD considers the diversity of both entities and relations simultaneously but with relatively low complexity. Thus, we choose to design our entity-level embedding learner based on TransD.
>
> We have added the following sentence to Section 4.3 to provide more explanations:
>
> "Translational models provide an intuitive solution by using the relation as a translation, enabling to explicitly model and constrain the learning of generalizable meta knowledge at the entity level."
>
> > **Question 2:** The pairwise interaction of the triples could have been done using any sort of permutation invariant operation, why SAB was picked?
>
> **Response to the question 2:** This is a natural and valid question. Our main goal when designing the MRL is to learn the commonality of multiple reference triplets associated with one same relation. There are two main considerations in our model design. (1) reference triplets are permutation invariant; (2) reference triplets that are more representative should be given higher weights when learning meta relation representation. For the purpose of taking both advantage of attention mechanism and permutation-invariant operations, we naturally picked SAB, since it can model the interactions between the set elements to learn meta relation representation.
>
> Although there are other permutation-invariant operations such as predefined heuristics such as max-pool, min-pool, sum, or average, as well as new learnable pooling methods [3, 4], most of these operations are however designed for aggregation purposes; for example, they are used as neighbor aggregators for learning node embeddings, or as pooling operators for aggregating graph-level representations. In addition, these operations process every element in a set independently, discarding interactions between elements [5]. Thus, they fail to fulfill our objective of learning generalizable meta relation representations through capturing pairwise interactions.
>
> We have added the following sentences to Section 4.2 to provide more justification:
>
> "There are two main considerations in our model design. (1) reference triplets are permutation-invariant; (2) reference triplets that are more representative should be given higher weights when learning meta relation representation."
>
> **Reference:**
>
> [3] Murphy, R., et al. "Janossy Pooling: Learning Deep Permutation-Invariant Functions for Variable-Size Inputs." International Conference on Learning Representations (ICLR 2019). 2019.
>
> [4] Grégoire Mialon, Dexiong Chen, Alexandre d’Aspremont, and Julien Mairal. A trainable optimal transport embedding for feature aggregation and its relationship to attention. In International Conference on Learning Representations, 2021.
>
> [5] Juho Lee, Yoonho Lee, Jungtaek Kim, Adam Kosiorek, Seungjin Choi, and Yee Whye Teh. Set transformer: A framework for attention-based permutation-invariant neural networks. In ICML, pp. 3744–3753, 2019.

---

### Official Review · Reviewer_Pexb · 2022-11-03

**Confidence:** 3
**Clarity, Quality, Novelty And Reproducibility:** 1. It is imperative to share the code…
**Correctness:** 3
**Technical Novelty And Significance:** 3
**Empirical Novelty And Significance:** 3
**Recommendation:** 6

**Strength And Weaknesses:**

The paper address an important problem in KBC. It identifies valid issues in existing methods. However:
1. Paper writing can be improved. Certain sections are hard to understand (especially sections 4.2/4.3/4.4). Since these sections form the core of the paper it is imperative to write them very clearly.
Adding details of MAML training startegy, MSA and SAB in appendix should be helpful.
2. \bigoplus used before definition
3. Some comments on the scalability of the model will be insightful.
4. Why did the authors not look at (?, r, t_j) queries along with (h_j, r, ?) and report the mean (the standard way of evaluation in KBC).
5. Definition 2: Consider adding an \exist r.
6. Some understanding of how the models perform on conventional KG completion datasets (where relations are associated with many more triples) is also important.
7. Authors should use a more competitive version of ComplEx (ComplEx-N3) for comparison. (See Lacroix, Timothée, Nicolas Usunier, and Guillaume Obozinski. "Canonical tensor decomposition for knowledge base completion." International Conference on Machine Learning. PMLR, 2018.)
Also why did the authors choose a translation model (sec 4.3) in place of more competitive KBC models like ComplEx?

**Summary Of The Paper:**

The paper proposes novel methods for few-shot KG completion. They identify two issues with existing methods for this task -- (a) They learn entity-level information from local nbr aggregators. (the paper jointly takes into account the nbr of head and tail entity for triple context) (b) They learn relation-level information using a sequential model (while the sequentiality assumption is invalid -- the paper use transformers instead). The authors proposed HiRe, which takes into account triplet-level interactions, context-level interactions, and entity-level information for query (h,r,?). MAML based training strategy used to train the model. The model shows improved performance on 2 benchmark datasets - Nell-One and Wiki-One. The ablation study demonstrates the importance of the key components of the model, where transformer-based MRL outshined.

**Summary Of The Review:**

I feel the paper has interesting contribution but I believe the quality of paper writing needs to be improved.

---

> ### Author Response · Authors · 2022-11-15
> **Response to Reviewer Pexb (Part 1/2)**
>
> We appreciate the reviewer's detailed and constructive feedback. We respond to your comments and provide clarifications below.
>
> > **Question1:** Paper writing can be improved. Certain sections are hard to understand (especially sections 4.2/4.3/4.4). Since these sections form the core of the paper it is imperative to write them very clearly. Adding details of MAML training strategy, MSA and SAB in appendix should be helpful. $\bigoplus$ used before definition. Definition 2: Consider adding an $\exists$ r.
>
> **Response to question 1:** We have carefully addressed your concerns on the writing of Section 4 and accordingly have updated it in the "rebuttal revision" of the paper (highlighted with blue font). In addition, the details of MSA, SAB and the MAML training strategy can be found in Appendix D1, D2 and E, respectively.
>
> > **Question 2:** Some comments on the scalability of the model will be insightful.
>
> **Response to question 2:** Thanks for your comments. We have performed a computational complexity analysis of our proposed HiRe. The following table shows the complexity in terms of the number of parameters and the number of multiplication operations per epoch, where $d$ denotes the dimension of entity embeddings, $k$ denotes the number of relation-entity tuples in the context, and $n$ denotes the number of reference triplets in each task. As can be seen, our proposed HiRe scales quadratically with the number of reference triplets in each task. Given the number of reference triplets is often very small ($\sim 5$) , our proposed HiRe scales reasonably well.
>
> |               |         \# Parameters         |           \# Operations           |
> | :-----------: | :---------------------------: | :-------------------------------: |
> | Context level |   $\mathcal{O}(nkd + nk^2)$   |   $\mathcal{O}(nkd^2 + nk^2d)$    |
> | Triplet level |    $\mathcal{O}(nd + n^2)$    |    $\mathcal{O}(nd^2 + n^2d)$     |
> | Entity level  |       $\mathcal{O}(d)$        |         $\mathcal{O}(d)$          |
> |     Total     | $\mathcal{O}(nkd + nk^2+n^2)$ | $\mathcal{O}(nkd^2 + nk^2d+n^2d)$ |
>
> We have added the above computational analysis in the revised paper. Please refer to Appendix H for details.
>
> > **Question 3:** Why did the authors not look at (?, r, $t_j$) queries along with ($h_j$, r, ?) and report the mean (the standard way of evaluation in KBC).
>
> **Response to question 3:** We would like to clarify that the widely adopted evaluation protocols for KG completion and few-shot KG completion are different. For KG completion, the standard way of evaluation is to perform negative sampling at runtime and report the mean ranks of missing head and tail entities. However, for few-shot KG completion, the evaluation is against a fixed candidate set provided for each missing tail entity.
>
> On the two widely used benchmark datasets: Nell-One and Wiki-One, released by [1], the candidate sets provided for evaluation are pre-specified as part of the datasets. Our work follows the same experimental protocol as all previous works on few-shot KGC, where the results on ($h_j$, r, ?) are reported for fair comparison.
>
> > **Question 4:** Some understanding of how the models perform on conventional KG completion datasets (where relations are associated with many more triples) is also important.
>
> **Response to question 4:** Thanks for the comments. We agree with the reviewer that more detailed understanding of model performance is important. Our results in Table 3 and Table 4 have already demonstrated that our model performs better when the number of reference triplets increases from 1-shot to 3-shot and 5-shot. This affirms greater advantages of our model when relations are associated with more training triplets. To further validate our model's capability with more reference triplets, we have conducted a new 10-shot experiment on Nell-One dataset. The results are as follows.
>
> |  HiRe   |  MRR  | Hits@10 | Hits@5 | Hits@1 |
> | :-----: | :---: | :-----: | :----: | :----: |
> | 1-shot  | 0.288 |  0.472  | 0.403  | 0.184  |
> | 3-shot  | 0.300 |  0.499  | 0.425  | 0.199  |
> | 5-shot  | 0.306 |  0.520  | 0.439  | 0.207  |
> | 10-shot | 0.333 |  0.526  | 0.449  | 0.231  |
>
> We can see that the performance of our model consistently improves with more training triplets on the few-shot settings. We expect this trend will continue with many more triplets. Unfortunately, the conventional KG completion datasets are organized in a very different form from few-shot KG completion datasets, making it difficult to directly evaluate our model on conventional KG completion datasets. Nevertheless, the above results provide a good indication of the performance of our model on conventional KG completion datasets.

---

> > ### Author Response · Authors · 2022-11-15
> > **Response to Reviewer Pexb (Part 2/2)**
> >
> > > **Question 5:** Authors should use a more competitive version of ComplEx (ComplEx-N3) for comparison. (See Lacroix, Timothée, Nicolas Usunier, and Guillaume Obozinski. "Canonical tensor decomposition for knowledge base completion." International Conference on Machine Learning. PMLR, 2018.)
> >
> > **Response to question 5:** Thanks for your suggestion. We have added ComplEx-N3 as a new baseline in our experiments. Please refer to Table 3 in the revised paper.
> >
> > The results of ComplEx-N3 and its comparison with our HiRe method on Nell-One dataset are provided below. From our results, we can see that our HiRe still consistently outperforms CompIEx-N3, although CompIEx-N3 offers improvements over CompIEx.
> >
> > It is worth noting that, due to the large scale of Wiki-One (consisting of over 4.8 million entities), training ComplEx-N3 on this dataset causes out of memory issue on all our available computing platforms (a single machine with 8 Tesla 32 GB V100 GPUs).
> >
> > | ComplEx-N3 |  MRR  | Hits@10 | Hits@5 | Hits@1 |
> > | :--------: | :---: | :-----: | :----: | :----: |
> > |   1-shot   | 0.206 |  0.335  | 0.271  | 0.140  |
> > |   5-shot   | 0.305 |  0.475  | 0.399  | 0.205  |
> > |  10-shot   | 0.319 |  0.477  | 0.406  | 0.230  |
> >
> > |  HiRe   |  MRR  | Hits@10 | Hits@5 | Hits@1 |
> > | :-----: | :---: | :-----: | :----: | :----: |
> > | 1-shot  | 0.288 |  0.472  | 0.403  | 0.184  |
> > | 5-shot  | 0.306 |  0.520  | 0.439  | 0.207  |
> > | 10-shot | 0.333 |  0.526  | 0.449  | 0.231  |
> >
> > > **Question 6:** Also why did the authors choose a translation model (sec 4.3) in place of more competitive KBC models like ComplEx?
> >
> > **Response to Question 6:**  Our choice of translation models (other than semantic matching models) is in line with the episode-based training paradigm [2] for few-shot learning problems. The training process is organized as a series of learning problems (episodes) so as to mimic the situation when unseen new tasks with very few samples arrive. Thus, few-shot models can learn to generalize from previously learned tasks to novel tasks.
> >
> > Specifically for few-shot KG completion, tasks are relation-specific. Thus, the key is to learn meta knowledge about given relations and generalize to unseen relations. As opposed to semantic matching models, translation models provide an intuitive solution and better fit our few-shot learning objective; using the relation as a translation enables to explicitly model and transfer generalizable meta knowledge about relations.
> >
> > We have added the following sentence to Section 4.3 to provide more explanations:
> >
> > "Translational models provide an intuitive solution by using the relation as a translation, enabling to explicitly model and constrain the learning of generalizable meta knowledge at the entity level."
> >
> > **Reference:**
> >
> > [1] Wenhan Xiong, Mo Yu, Shiyu Chang, Xiaoxiao Guo, and William Yang Wang. One-shot relational learning for knowledge graphs. In EMNLP, pp. 1980–1990, 2018.
> >
> > [2] Vinyals, Oriol, et al. "Matching networks for one shot learning." Advances in neural information processing systems 29 (2016).

---

### Author Response · Authors · 2022-11-15
**Response to Reviewers**

Dear Reviewers,

We are grateful to all reviewers for your many constructive comments and helpful feedback. To address your main concerns, we have done our best to improve our work. Our point-by-point responses can be found below, with pointers to locations where they have been addressed or modified in the revised paper.
We have also updated our paper based on your comments accordingly and highlighted the differences with previous version in blue.
We appreciate all the suggestions made by reviewers to improve our work. It is our pleasure to hear your feedback and we look forward to answering your follow-up questions.

Paper315 Authors

---

### Decision · Program_Chairs · 2023-01-20

**Decision:**

Accept: poster

**Justification For Why Not Higher Score:**

Incremental novelty

**Justification For Why Not Lower Score:**

Strong results, good analysis.

**Metareview: Summary, Strengths And Weaknesses:**

The paper presents a new method for few shot KGC, i.e., KGC for relations in the long tail. The main idea is to use shared context between head and tail entity neighborhoods in  embedding the triple via contrastive learning. In particular, three capture three levels of relational information: a. Context level b. Triplet level c. Entity level. They call this "hierarchical relational learning", although thats a bit of misnomer.

Main strengths are (1) careful analysis of research gaps in existing works, which result in this solution, (2) extensive ablation study, and (3) impressive experimental results on one of the two datasets; good results on the 2nd.

Main weaknesses are (1) incremental novelty: it appears that most ideas are present in other works, this particular combination is not. (2) name hierarchical learning or hierarchical relation learning is a misnomer -- since it is neither doing hierarchical learning not learning hierarchy of relations.

On balance, the paper can be accepted. Incremental novelty is considered an issue with most papers these days, since coming up with a new building block has become harder and harder. This paper has several other advantages as discussed above. Three of the reviewers (8,6,5) did not respond to author comments. One reviewer (3) increased their score to 5. Authors have diligently answered most questions.

**Note From Pc:**

if the above contains the word "oral" or "spotlight" please see: "oral" presentation means -> notable-top-5% and "spotlight" means -> notable-top-25%. As stated in our emails, we are disassociating presentation type from AC recommendations